

# SELEN⁴ (SELEN version 4.0): a Fortran program for solving the gravitationally and topographically self-consistent Sea Level Equation in Glacial Isostatic Adjustment modeling

Giorgio Spada[a] and Daniele Melini[b]

[a] Dipartimento di Scienze Pure e Applicate (DiSPeA), Università di Urbino "Carlo Bo", Italy
[b] Istituto Nazionale di Geofisica e Vulcanologia, Via di Vigna Murata 605, I-00143 Rome, Italy

**Correspondence:** Giorgio Spada (*giorgio.spada@gmail.com*)

**Abstract.** We present SELEN⁴ (a SealEveL EquatioN solver), an open-source program written in Fortran 90 that simulates the Glacial Isostatic Adjustment (GIA) process in response to the melting of the late-Pleistocene ice sheets. Using a pseudo-spectral approach complemented by a spatial discretization on an icosahedron-based spherical geodesic grid, SELEN⁴ solves a

generalised "Sea Level Equation" (SLE) for a spherically symmetric Earth with linear viscoelastic rheology, taking the migration of the shorelines and the rotational feedback on sea level into account. The approach is gravitationally and topographically self-consistent, since it considers the gravitational interactions between the solid Earth, the cryosphere and the oceans, and it accounts for the evolution of the Earth's topography in response to changes in sea level. Program SELEN⁴ can be employed to study a broad range of geophysical effects of GIA, including past relative sea-level variations induced by the melting of the

late-Pleistocene ice sheets, the time-evolution of paleogeography and of the ocean function since the Last Glacial Maximum, the history of the Earth's rotational variations, present-day geodetic signals observed by Global Navigation Satellite Systems and geopotential field variations detected by satellite gravity missions like GRACE (the Gravity Recovery and Climate Experiment). The *GIA fingerprints* constitute a standard output of SELEN⁴. Along with the source code, we provide a supplementary document with a full account of the theory, some numerical results obtained from a standard run, and a User guide. Pro-

gram `SELEN` was conceived by GS in 2005 as a tool for students eager to learn about GIA. Still, it is the only open-source program for the solution of the SLE available to the community.

## 1 Introduction

In the last few decades, GIA modeling has progressively gained a central role in the study of contemporary sea-level change. Sea-level variations observed at the tide gauges deployed along the world coastlines need to be decontaminated from the effect

of GIA to enlighten the effects of global warming. As discussed in the review of Spada and Galassi (2012), a precise estimate of global sea-level rise has been possible only after Peltier and Tushingham (1989) first solved the SLE using an appropriate spatial resolution, building upon the seminal papers of Farrell and Clark (1976) and Clark et al. (1978). Since then, a number of





GIA models characterised by different assumptions about the Earth's rheological profile and the history of the late-Pleistocene ice sheets have been proposed, constrained by sea-level proxies available since the Last Glacial Maximum (for a review of the development of GIA modelling, see Whitehouse, 2009; Spada, 2017; Whitehouse, 2018). These models have provided increasingly accurate estimates of global mean secular sea-level rise (a summary is given in Table 1 of Spada and Galassi,

2012), but have also the potential of describing the patterns of future trends of sea level in a global change scenario (see *e.g.,* Bamber et al., 2009; Spada et al., 2013). Since the beginning of the "altimetry era" (1992-today) and the launch of the Gravity Recovery and Climate Experiment (GRACE, see Wahr et al., 1998) in 2002, GIA modeling has re-gained momentum, providing the tools for isolating the effects of global warming *i)* from absolute sea-level data (Nerem et al., 2010; Cazenave and Llovel, 2010) and *ii)* from the Stokes coefficients of the gravity field (see Leuliette and Miller, 2009; Cazenave et al.,

2009; Chambers et al., 2010; WCRP, 2018) to infer the ocean mass variation. Despite GIA is now tightly integrated into the science of global change (Church et al., 2013b), little efforts have been payed so far to the development of open-source codes for the solution of the SLE, although several Post Glacial Rebound simulators (like *e.g.,* TABOO, see Spada et al., 2004, 2011) and Love numbers calculators have been made available to the community (Spada, 2008; Melini et al., 2015; Bevis et al., 2016; Kachuck and Cathles, 2019). As far as we know, the only publicly available and open-source SLE solver in which the

viscoelastic rheology of the mantle is properly taken into account is SELEN. The SLE solver ISSM-SESAW v1.0 of Adhikari et al. (2016), being oriented to short term cryosphere and climate climate changes, is limited the elastic rheology.

SELEN was first presented to the GIA modeling community by Spada and Stocchi (2007), who numerically implemented the SLE theory reviewed in Spada and Stocchi (2006). SELEN was fully based on the classical formulation of Farrell and Clark (1976); hence, the fixed-shorelines approximation was assumed, and no account was given of rotational effects on sea-

level variations. SELEN used the Love numbers calculator TABOO (see Spada et al., 2011) as a subroutine and was tied to the Generic Mapping Tools (GMT, see Wessel and Smith, 1998) for the construction of the present-day ocean function. In SELEN and in all its subsequent versions, the numerical integration of the SLE over the sphere takes advantage of the icosahedron-based pixelization by Tegmark (1996). Similarly, all the versions are based upon the pseudo-spectral method of Mitrovica and Peltier (1991) and Mitrovica et al. (1994) for the solution of the SLE. Originally, SELEN came without a User guide, and it was

disseminated via email by the authors. After SELEN was first published in 2007, a number of improvements were made in terms of computational efficiency, portability and versatility, but leaving the physical ingredients of the original code unaltered. This led to a new version of the program, named SELEN 2.9 and announced by Spada et al. (2012). Since 2015, the Computational Infrastructure of Geodynamics (CIG, *http://www.geodynamics.org/*) is hosting SELEN 2.9 in the Short-term Crustal Dynamics section of its page (see *http://geodynamics.org/cig/software/selen*), from where it can be freely downloaded along with a theory

booklet and a fully detailed User guide (Spada and Melini, 2015). Since year 2012, with the aid of Florence Colleoni and thanks to the feedback of a number of colleagues and students, GS and DM have implemented new modules aimed at solving the SLE in the presence of rotational effects and taking the migration of the shorelines into account. This has progressively led to several *interim* versions of the program (SELEN 3.*x*), which have been tested intensively and validated during the years, but never officially released. We note that building upon SELEN, some colleagues have independently developed other versions of

the code aimed at specific tasks, such as the study of the coupling between the SLE and ice dynamics (de Boer et al., 2017).





Taking advantage of the experience developed since `SELEN` was first designed, we are now publishing a new version of the code named SELEN[4]. With respect to previous versions, SELEN[4] has been improved in several aspects. *i)* The underlying SLE theory has been fully revised and now accounts both for horizontal migration of shorelines and for rotational effects, resulting in a more realistic description of the GIA processes. *ii)* The package has been streamlined and reorganized into two

independent modules: a *solver*, which obtains a numerical solution of the SLE in the spectral domain, and a *post-processor*, which computes a full suite of observable quantities through a spherical harmonic synthesis. This new structure facilitates code portability, reusability and customization, enabling the adaptation of SELEN[4] to new use-cases. *iii)* The SELEN[4] modules have been completely rewritten using symbol names that are closely matching those of the variables introduced in this paper, for the ease of code readability. Particular attention has been paid to the optimization of the SLE solver, resulting in a large extent

of shared-memory parallelism, which allows for an efficient scaling to high resolutions on multi-core systems. *iv)* SELEN[4] has been decoupled from the GMT software package, which is no longer strictly required to run a GIA simulation, thus facilitating code portability on high-performance systems where GMT may not be available. SELEN[4] still takes advantage of GMT (version 4) to produce various graphical outputs through plotting scripts included in the distribution package. *v)* SELEN[4] no longer calls the post glacial rebound solver `TABOO` as an internal subroutine to compute the viscoelastic loading and tidal

Love numbers, which are instead supplied by the user through a data file. In this way, any set of Love numbers can be used in SELEN[4], possibly overcoming some of the intrinsic limitations of existing Love numbers calculators like `TABOO`, *vi)* Recently, a prototype version of SELEN[4] has been successfully validated in a community benchmark of independently developed GIA codes (Martinec et al., 2018).

The paper is organized into three main sections. In the first one, we present a condensed theory background for the SLE,

leaving the details to the supplement. In the second, we describe the outputs obtained by a standard, intermediate-resolution run of SELEN[4]. In the third section, we draw our conclusions.

## 2 Theory

Here we obtain the sea-level equation (SLE) from first principles, leaving a number of details of the theory to the supplementary material, hereafter referred to as SSM19. The focus is on the various forms taken by the SLE, which are characterised by an

increasing complexity; the goal is to obtain a formulation suitable for a numerical discretisation, which is given and analysed in SSM19 and implemented in SELEN[4]. In an attempt to simplify the presentation, and to obtain compact expressions for all the quantities involved, we are not exactly following the traditional notation adopted in the literature since the seminal paper by Farrell and Clark (1976), hereafter referred to as FC76. The number of definitions and variables involved in the construction of the SLE is remarkable and some derivations are cumbersome; to facilitate the readers – especially those who are approaching

the GIA problem for the first time – we refer to the synopsis presented in SSM19. Program SELEN[4] is written in a plain way, adopting constants and variables names that follow the same notation employed in this theory section and in SSM19, with the aim of facilitating code readability.





## 2.1 Surface loads

We consider the system composed by the ice and by the water in the oceans, at a given time $t$. Its mass can be expressed as

$$M(t) = \int_e L \, dA, \tag{1}$$

where $L(\gamma, t)$ is the *surface load*, $\gamma$ stands for $(\theta, \lambda)$ where $\theta$ and $\lambda$ are the geocentric colatitude and longitude, respectively, the

integral is over the whole Earth's surface, and $dA = a^2 \sin\theta d\theta d\lambda$ is the area element, $a$ being the average radius of the Earth. According to Eq. (1), $L$ represents the mass per unit area distributed over the Earth's surface. In SSM19, we show that $L$ can be expressed as the sum of two contributions, which account *i)* for the load exerted by the grounded ice and *ii)* for the load of water on the oceans floors, respectively:

$$L(\gamma, t) = \rho^i IC + \rho^w BO, \tag{2}$$

where $I$ is the *ice thickness*, $\rho^i$ and $\rho^w$ are the ice and ocean water densities, $O$ is the *ocean function* (OF), $C = 1 - O$ is the *continent function* (CF), and

$$B(\gamma, t) = -T \tag{3}$$

is *sea level*, where $T$ is bedrock *topography* (*e.g.,* Kendall et al., 2005). Note that due to the horizontal migration of the shorelines and to the transition between floating and grounded ice, the OF and the CF are, in general, time-dependent.

In the following, we are concerned with *time variations* of the fields involved in the SLE, referred to a reference state established for $t \leq 0$. Accordingly, using Eq. (2) in (1), the *mass variation* $\mathcal{M}(t) = M - M_0$ of the system composed by ice and water is

$$\mathcal{M}(t) = \rho^i \int_e (IC - I_0 C_0) \, dA + \rho^w \int_e (BO - B_0 O_0) \, dA, \tag{4}$$

where subscript $0$ denotes reference quantities, and the first term on the right hand side represents the mass variation of the

grounded ice, which we denote by $\mu(t)$. Since mass must be conserved, we have

$$\mathcal{M}(t) = \int_e \mathcal{L} \, dA = 0, \tag{5}$$

where

$$\mathcal{L}(\gamma, t) \equiv L - L_0. \tag{6}$$

is the *surface load variation*. As shown in SSM19, equivalent forms of the mass conservation constraint (5) are

$$<\mathcal{L}>^e (t) = 0 \tag{7}$$





where $< ... >^e$ indicates the average over the whole Earth surface, and

$$\mathcal{L}_{00}(t) = 0, \tag{8}$$

where $\mathcal{L}_{00}$ is the degree $l = 0$ and order $m = 0$ coefficient in the spherical harmonics expansion of $\mathcal{L}(\gamma, t)$. Hereinafter, we shall consider only *plausible surface loads*, for which the mass is conserved (see Bevis et al., 2016, for a discussion).

In SSM19, a suitable decomposition is found for $\mathcal{L}$, namely

$$\mathcal{L}(\gamma, t) = \mathcal{L}^a + \mathcal{L}^b + \mathcal{L}^c, \tag{9}$$

where the first term

$$\mathcal{L}^a(\gamma, t) = \rho^i C \mathcal{I} \tag{10}$$

is associated with the *ice thickness variation*

$$\mathcal{I}(\gamma, t) = I - I_0, \tag{11}$$

the second

$$\mathcal{L}^b(\gamma, t) = \rho^w O \mathcal{S} \tag{12}$$

stems from *sea-level change*

$$\mathcal{S}(\gamma, t) = B - B_0, \tag{13}$$

and the third

$$\mathcal{L}^c(\gamma, t) = \rho^r Q \mathcal{O}, \tag{14}$$

is associated with *OF variations*

$$\mathcal{O}(\gamma, t) = O - O_0, \tag{15}$$

where $\rho^r$ is an arbitrary reference density and $Q$ a time invariant auxiliary variable.

**2.2   The Sea Level Equation**

Above, *sea level* has been defined as $B = -T$, where $T$ is topography (see Eq. 3). Denoting by $r^{ss}(\gamma, t)$ and $r^{se}(\gamma, t)$ the radii of the sea surface and of the Earth's solid surface in a geocentric reference frame, respectively, sea level can be equivalently expressed as

$$B(\gamma, t) = r^{ss} - r^{se}, \tag{16}$$





and in the reference state

$$B_0(\gamma) = r_0^{ss} - r_0^{se}. \tag{17}$$

So, introducing the *sea surface variation*

$$\mathcal{N}(\gamma, t) = r^{ss} - r_0^{ss} \tag{18}$$

and the *vertical displacement* of the solid surface of the Earth

$$\mathcal{U}(\gamma, t) = r^{ss} - r_0^{ss}, \tag{19}$$

using Eq. (13), we obtain the SLE in its most basic form

$$\mathcal{S}(\gamma, t) = \mathcal{N} - \mathcal{U}. \tag{20}$$

The sea surface variation is tightly associated with variations in the Earth's gravity field. Indeed, FC76 have shown that

$$\mathcal{N}(\gamma, t) = \mathcal{G} + c, \tag{21}$$

where

$$\mathcal{G}(\gamma, t) = \frac{\Phi}{g}, \tag{22}$$

is the *displacement of the geoid*, and $\Phi$ is the total *variation of the geopotential* (including surface loading and rotational effects), $g$ is the reference gravity acceleration evaluated at the Earth's surface, and $c$ is a spatially invariant term that shall be

determined imposing the constraint of mass conservation (see also Tamisiea, 2011; Spada, 2017). Hence the SLE (20) becomes

$$\mathcal{S}(\gamma, t) = \mathcal{R} + c, \tag{23}$$

where

$$\mathcal{R}(\gamma, t) = \mathcal{G} - \mathcal{U}, \tag{24}$$

shall be referred to as *sea-level response function*.

We now assume that the responses to surface loading and to changes in the centrifugal potential can be combined linearly. Accordingly, the SLE (23) can be further rearranged as

$$\mathcal{S}(\gamma, t) = \mathcal{R}^{sur} + c + \mathcal{R}^{rot}, \tag{25}$$

where

$$\mathcal{R}^{sur}(\gamma, t) = \mathcal{G}^{sur} - \mathcal{U}^{sur} \tag{26}$$





and

$$\mathcal{R}^{rot}(\gamma,t) = \mathcal{G}^{rot} - \mathcal{U}^{rot} \tag{27}$$

are the *surface* and the *rotation sea-level response functions*, whereas

$$\mathcal{G}(\gamma,t) = \mathcal{G}^{sur} + \mathcal{G}^{rot} \tag{28}$$

and

$$\mathcal{U}(\gamma,t) = \mathcal{U}^{sur} + \mathcal{U}^{rot} \tag{29}$$

are the *geoid* and the *vertical displacement response functions*, respectively.

By the constraint of mass conservation given by Eq. (5), the $c$ constant is easily determined. Using for $c$ the expression found in SSM19, the SLE (25) becomes

$$\mathcal{S}(\gamma,t) = \mathcal{S}^{ave} + \left(\mathcal{R}^{sur} - <\mathcal{R}^{sur}>^o\right) + \left(\mathcal{R}^{rot} - <\mathcal{R}^{rot}>^o\right), \tag{30}$$

where $<\dots>^o$ indicates the average over the (time-dependent) ocean surface defined by $O = 1$, and

$$\mathcal{S}^{ave}(t) = \mathcal{S}^{equ} + \mathcal{S}^{ofu}, \tag{31}$$

where $\mathcal{S}^{equ}$ and $\mathcal{S}^{ofu}$ are two spatially invariant terms. The first, referred to as *equivalent sea-level change*, is

$$\mathcal{S}^{equ}(t) \equiv -\frac{\mu}{\rho^w A^o}, \tag{32}$$

where $\mu$ is the mass variation of the grounded ice and $A^o$ is the area of the oceans. The second

$$\mathcal{S}^{ofu}(t) \equiv \frac{1}{A^o}\int_e T_0 \mathcal{O} dA \tag{33}$$

depends explicitly upon variations of the OF, either due to the horizontal migration of the shorelines or to transitions from grounded to floating ice (or *vice versa*). Evaluating the ocean-average of both sides of Eq. (30), and observing that $<<\mathcal{R}>^o>^o = <\mathcal{R}>^o$ and that $<\mathcal{S}^{ave}>^o = \mathcal{S}^{ave}$, it is easily verified that $\mathcal{S}^{ave}$ simply represents the ocean-averaged relative

sea-level change

$$\mathcal{S}^{ave}(t) \equiv <\mathcal{S}>^o. \tag{34}$$

In consequence of that, from Eq. (30) we see that the regional imprint of GIA on relative sea-level is totally determined by the response functions $\mathcal{R}^{sur}$ and $\mathcal{R}^{rot}$.

In the classical FC76 framework, a constant OF is assumed and the effects arising from Earth rotation are neglected. Hence

$\mathcal{R}^{rot} = 0$ and $\mathcal{O} = 0$, with the latter condition implying $\mathcal{S}^{ofu} = 0$ because of (33). In this context, the SLE simply reduces to

$$\mathcal{S}^{FC76}(\gamma,t) = \mathcal{S}^{eus} + \left(\mathcal{R}^{sur} - <\mathcal{R}^{sur}>^o\right), \tag{35}$$





where

$$\mathcal{S}^{eus}(t) = -\frac{\mu}{\rho^w A^{op}} \tag{36}$$

defines *eustatic sea-level change*, and $A^{op}$ represents the present-day area of the surface of the oceans. Hence, for a rigid and non-gravitating Earth (for which $\mathcal{R}^{sur} = 0$), $\mathcal{S}^{eus}$ would represent the (spatially invariant) relative sea-level change. Note that

$\mathcal{S}^{eus}$ should not be confused with $\mathcal{S}^{ave}$ given by Eq. (31) since the latter is dynamically dependent upon the Earth's response through $A^o(t)$ and $\mathcal{O}(t)$ (*e.g.,* Spada, 2017).

Following *e.g.,* Milne and Mitrovica (1998), the surface response function $\mathcal{R}^{sur}$ in (30) is obtained by a 3-D *spatio-temporal* convolution

$$\mathcal{R}^{sur}(\gamma, t) = \Gamma^s \otimes \mathcal{L}, \tag{37}$$

where $\Gamma^s(\gamma, t)$ the *surface sea-level Green's function*. The details of the expansion of $\mathcal{R}^{sur}$ in series of spherical harmonics are somewhat cumbersome and are left to SSM19. Here we only note that using Eq. (9) we have $\mathcal{R}^{sur} = \mathcal{R}^a + \mathcal{R}^b + \mathcal{R}^c$, where the three terms are obtained by convolving $\Gamma^s$ with $\mathcal{L}^a$, $\mathcal{L}^b$ and $\mathcal{L}^c$, respectively. Contrary to $\mathcal{R}^{sur}$, the harmonic coefficients of the rotational response $\mathcal{R}^{rot}(\gamma, t)$ are directly obtained by a 1-D *time* convolution

$$\mathcal{R}^{rot}_{lm}(t) = \Upsilon^s_l * \Lambda_{lm}, \tag{38}$$

where $\Upsilon^s_l(t)$ is the *rotation sea-level Green's function* and $\Lambda_{lm}(t)$ are the coefficients of degree $l$ and order $m$ of the expansion of the *variation of centrifugal potential* $\Lambda(\gamma, t)$ associated with changes in the Earth's angular velocity (Milne and Mitrovica, 1998). In SSM19, it is shown that $\Lambda(\gamma, t)$ is essentially a spherical harmonic function of degree $l = 2$ and order $m = \pm 1$.

Thus, Eq. (30) can be rearranged as

$$\mathcal{S}(\gamma, t) = \mathcal{S}^{ave} + \mathcal{R}'^{\,a} + \mathcal{R}'^{\,b} + \mathcal{R}'^{\,c} + \mathcal{R}'^{\,rot}, \tag{39}$$

where the primed response functions are

$$\mathcal{R}'^{\,abc}(\gamma, t) = \mathcal{R}^{abc} - <\mathcal{R}^{abc}>^o, \tag{40}$$

$$\mathcal{R}'^{\,rot}(\gamma, t) = \mathcal{R}^{rot} - <\mathcal{R}^{rot}>^o. \tag{41}$$

We note that $\mathcal{R}'^{\,b}$ depends on $O\mathcal{S}$ through the surface load variation $\mathcal{L}^b$ (see Eq. 12); in SSM19 it is shown that this holds for $\mathcal{R}^{rot}$ as well. Following Mitrovica and Peltier (1991), in view of the numerical solution of the SLE it is therefore convenient

to transform Eq. (39) in such a way that $\mathcal{Z} = O\mathcal{S}$ becomes the unknown *in lieu* of $\mathcal{S}$. This is accomplished projecting Eq. (39) on the OF (*i.e.,* multiplying both sides of the SLE by $O$), which provides the final form of the SLE

$$\mathcal{Z}(\gamma, t) = \mathcal{Z}^{ave} + \mathcal{K}^a + \mathcal{K}^b(\mathcal{Z}) + \mathcal{K}^c + \mathcal{K}^{rot}(\mathcal{Z}), \tag{42}$$

where

$$\mathcal{Z}(\gamma, t) = O\mathcal{S} \tag{43}$$





and

$$\mathcal{Z}^{ave}(\gamma, t) = O\mathcal{S}^{ave} = O\left(\mathcal{S}^{equ} + \mathcal{S}^{ofu}\right) \tag{44}$$

$$\mathcal{K}^{abc}(\gamma, t) = O\mathcal{R}'^{\,abc} \tag{45}$$

$$\mathcal{K}^{rot}(\gamma, t) = O\mathcal{R}'^{\,rot}. \tag{46}$$

The dependence of $\mathcal{K}^b$ and $\mathcal{K}^{rot}$ upon $\mathcal{Z}$ in Eq. (42) manifests the implicit nature of the SLE, which is a *3-D non-linear integral equation*, similar, in some respect, to an inhomogeneous and non linear Fredholm equation of the second kind (*e.g.,* Jerri, 1999; Spada, 2017). For the spectral discretization of Eq. (42) and for an illustration of the iterative scheme adopted to solve the SLE, the reader is referred to sections S7 and S8.7 of SSM19.

## 3    A test run with SELEN[4]

In the following, we illustrate some of the outputs of a standard SELEN[4] run in which the resolution of the Tegmark grid is set to $R = 44$ (see S8.6), the maximum harmonic degree of the spectral decomposition[1] is $l_{max} = 128$ and we solve the SLE by three external and three internal iterations ($n_{ext} = n_{int} = 3$, see S8.7). Note that three iterations are normally adopted in GIA studies, like in Kendall et al. 2005. Henceforth, the notation *R44/L128/I3* shall be employed to denote these fundamental SELEN[4] settings. Of course, with increasing values of parameters ($R$, $l_{max}$, $n_{ext}$, $n_{int}$), more accurate results are expected,

which however might come with a substantial increase in the computational burden.

We assume that the user has installed and executed the program following the guidelines on the User guide of SELEN[4]. Most of the program outputs discussed in this section have been obtained using the same configuration file that comes with the SELEN[4] package. However, some results shall be based on different settings, in order to appreciate the sensitivity of the outputs on key configuration parameters. We first describe the GIA model adopted in the test run, which consists of three elements,

*i.e.,* a ice melting history, a description of the present-day global relief, and a 1-D rheological model of the Earth's mantle. Then, browsing output folders of SELEN[4], we illustrate and discuss two distinct output sets, pertaining to the past and to the present effects of GIA on sea-level change and on geodetic variations, respectively. The features of the test run are summarized in Table 1. For reference, on a 12-core Mid-2012 Mac Pro, the execution time of this test run of SELEN[4] is 1h and 15 min.

### 3.1    GIA model

In principle, there are no restrictions on the spatial and temporal features of the ice melting history that can be employed in SELEN[4], provided that the model is properly discretized according to the scheme outlined in S8.7. Similarly, any linear rheological profile is *a priori* acceptable for the mantle and for the lithosphere, as long as the Love numbers can be cast in a normal-mode multi-exponential form (Peltier, 1974). Due to its central role in the context of contemporary GIA studies, in

---

[1]Note that condition $P \ge l_{max}^2/3$, where $P = 40R(R-1) + 12$ is the number of pixels in the grid for a given $R$ value, must be necessarily met to preserve the properties of the spherical harmonics on the grid. See SSM19 and Tegmark (1996) for details.





our test run we have implemented an *ad hoc* realization of the GIA model ICE-6G_C(VM5a), originally introduced by Peltier et al. (2015).

### 3.1.1 Ice melting history

Ice thickness data for model ICE-6G_C have been downloaded from the home page of Prof. WR Peltier on August 2016.

The data span the last $26,000$ yrs and are provided on a $1° \times 1°$ global cartesian latitude-longitude grid (the number of grid points is thus $64,800$). In each of the grid cells, the time history of ice thickness is assumed to evolve in a piecewise linear manner, with variable increments of $1.0$ or $0.5$ kyrs. Thus, to fit the SELEN[4] default input format, we have first re-mapped the original thickness data on a spherical equal-area Tegmark grid described in S8.6 of SSM19. In doing that, we have chosen a resolution parameter $R = 44$, so that the grid consists of $P = 75,692$ pixels (or cells), each with a radius of $\sim 46$ km. The cells

number is thus comparable to the number of cells in the original cartesian grid ($64,800$). In addition, we have transformed the original time history in a piecewise constant form with a uniform spacing of $0.5$ kyrs, assuming no glaciation phase prior to deglaciation. Because of the adaptations we have made, the ice model so obtained is not an exact replica of ICE-6G_C, but a particular *realisation* of it. Hence, to avoid any ambiguity, in the following it shall be referred to as I6G-T05-R44. Assuming an ice density $\rho^w = 931$ kg m$^{-3}$ and that the area of the oceans is fixed to the present value, I6G-T05-R44 holds $206.5$ m of

equivalent sea level at $26$ ka and $75.1$ m at present, corresponding to a total eustatic sea-level rise of $131.4$ m since the inception of melting ($26$ ka). The ice thickness of I6G-T05-R44 for a few time frames is shown in Figure 1.

### 3.1.2 Present day topography

In order to reconstruct the whole history of the Earth's topography and of Relative Sea Level (RSL) since the inception of deglaciation, it is necessary to impose the present relief as a *final condition* for the Sea Level Equation (see Peltier, 1994).

In this test run, we have utilized the "bedrock version" of the global dataset ETOPO1 (Amante and Eakins, 2009; Eakins and Sharman, 2012) as the final condition, whereas the final condition for the ice thickness is given by the last time frame of I6G-T05-R44. ETOPO1 is distributed on a cartesian longitude-latitude grid with a resolution of $1$ arc-minute ($\frac{1}{60}°$), so that an interpolation on the Tegmark grid is necessary in SELEN[4] before it can be utilized. Other choices of the final topography are of course possible. For example, in order to ensure the maximum accuracy in Antarctica, the original model ICE-6G_C employs

the *Bedmap2* data set of Fretwell et al. (2013) south of $60°$S latitude. In SELEN[4], there are no restrictions on the choice of the final relief, which is left to the user. In Figure 2 we show the realization of the modern bathymetry that we have obtained by interpolating ETOPO1 on the same Tegmark grid that we have used for the ice sheets in this test run, which shall be referred to as model ETO-R44 in the following. With SELEN[4], other versions of this elevation model are made available, characterized by different spatial resolutions $R$.





### 3.1.3 Rheological profile

The 11-layer Maxwell rheological profile of the 1-D Earth employed in the test run is shown in Table 2. For each layer, values of density and of rigidity are obtained by volume-averaging the PREM (Preliminary Reference Earth Model) of Dziewonski and Anderson (1981) while the viscosity profile is reproduced using the data available in the supporting information supplied

with Peltier et al. (2015). The 90-km thick lithosphere is elastic and the core is fluid, homogeneous and inviscid. Note that since the original VM5a profile includes elastic compressibility and reproduces the finely layered PREM structure (Peltier et al., 2015), in the following we shall refer to the model in Table 2 as VM5i. As it includes $N_v = 9$ Maxwell layers in the mantle, characterized by distinct properties, for any given harmonic degree $l$, the loading Love numbers (LLNs) and tidal Love numbers (TLNs) for model VM5i are described by a spectrum of $4\,N_v = 36$ viscoelastic normal modes (see *e.g.,* Spada et al.,

2011). Since agreed results on the Love numbers of a multi-layered compressible viscoelastic model have not been obtained yet, in the test run we rest on the incompressible profile VM5i. We remark, however, that SELEN[4] can work with compressible or transient rheologies as well, provided that LLNs and TLNs in normal-mode form (Wu and Peltier, 1982) are accessible to the user. Figures 3a and 3b show the elastic and fluid values of the LLNs in the range of harmonic degrees $1 \le l \le 1,024$ for model VM5i. The Love numbers are given in a geocentric reference frame with origin in the whole Earth's center of mass

(CM). For reference, numerical values of a few relevant LLNs and TLNs are listed in Table 3. The LLNs and TLNs have been computed by the Love numbers calculator `TABOO` (see Spada et al., 2011) in a multi-precision environment (Spada, 2008).

### 3.2 Glacial Isostatic Adjustment in the past

This section is devoted to the description of some outputs of the SELEN[4] test run, concerning the effects of GIA during the whole period after the LGM. These include *i)* the predictions of the history of relative sea level (RSL) at specific sites, *ii)* the

time-evolution of paleo-topography in some regions of interest, and *iii)* the excursions of the Earth's pole of rotation forced by GIA.

### 3.2.1 Relative Sea Level curves

Figure 4 shows data (with error bars) and SELEN[4] predictions for a small subset of the 392 sites contained in the RSL database of Tushingham and Peltier (1993) (hereafter referred to as TP93). In view of its historical importance in the devel-

opment of GIA studies (*e.g.,* Tushingham and Peltier, 1991, 1992; Melini and Spada, 2019), the TP93 database is available with the SELEN[4] package; however, there are no restrictions on the use of other datasets, or simply individual RSL records, if available to the user. The black curves in Figure 4 have been obtained using the GIA model described in Section 3.1, characterized by the settings *R44/L128/I3*. Blue curves have been obtained by configuring SELEN[4] with the combination of parameters *R100/L512/I5 i.e.,* increasing the spatial resolution and the number of internal and external iterations in a significant way (a

truncation degree $l_{max} = 256$ is often employed in GIA modeling, see *e.g.,* Kendall et al. 2005). With this configuration, the pixel radius is reduced to $\sim 20$ km, see S8.6. Of course, the execution time of SELEN[4] increases significantly with respect to the test run, requiring 2.5 days ($\sim 60$ h) on a 56-core Intel Xeon E7 "Broadwell" system. It is apparent that this high-resolution





case is providing results that substantially match those of the standard run with *R44/L128/I3*. Minor differences can be noted in the early stages of deglaciation, which however do not exceed the typical uncertainty on the observed RSL values. These differences are likely to be caused by the significant changes that the topography undergoes in this early phase in the polar regions, which are better captured by increasing the model resolution. Finally, red curves have been obtained for a low resolution run

with *R30/L64/I2*, whose execution time is 15 min on a 12-core Mid-2012 Mac Pro. The curves clearly indicate that computationally inexpensive runs can provide reliable results in the far field of the previously glaciated areas (*e.g.,* in sites 639 and 535), but in the near field (*e.g.,* site 238) they can diverge significantly from high-resolution but also from intermediate-resolution results.

From a visual inspection of Figure 4, it is apparent that at some sites the best GIA predictions fit very well the observations,

like for sites 101 and 283. For others, the trend of the RSL data is captured satisfactorily (see sites 155, 209 and 328) while in some others the fit is quite poor (sites 639, 525 and 570). The identification of the possible sources of the evidenced misfits, which should be measured using rigorous statistical methods, is not the purpose of this work. We only note that they do not necessarily stem from limitations of the GIA model adopted, since it is well known that at a specific site tectonic deformations can have important roles (see *e.g.,* Antonioli et al., 2009, for a significant example) and these are not taken into account when

solving the SLE. Similarly, in our formulation of the SLE we are neglecting the possible effects from the loads exerted by sediments (Dalca et al., 2013). Since program SELEN[4] is open source, the users can modify the code to account for non glacial loads and change the configuration to determine more suitable combinations of the basic ingredients of GIA modeling *i.e.,* the history of deglaciation and the rheological layering of the mantle, in order to improve the fit between model predictions and any preferred dataset.

**3.2.2  Paleo-topography**

Differently from previous versions of the program, SELEN[4] allows for a *gravitationally* and *topographically* self-consistent description of the evolution of sea level, along the route highlighted by Peltier (1994) and Lambeck (2004). This implies that SELEN[4] can iteratively reconstruct the time evolution of the coastlines and of the OF, in a fashion that is consistent with the gravitational, rotational and deformational effects induced by deglaciation (for a full account of the theory, the reader is referred

to SSM19). These features make the SLE an integral 3-D non-linear equation (Spada, 2017). The importance of the evolution of paleo-topography for the development of human culture since the LGM has been pointed out in a number of works (see *e.g.,* Cavalli-Sforza et al., 1993; Peltier, 1994; Lambeck, 2004; Dobson, 2014, and references therein). Recently, in the context of GIA modeling,  Spada and Galassi (2017) have faced the problem of the dynamic evolution of *aquaterra*, *i.e.,* the land that has been inundated and exposed during the last glacial cycle (Dobson, 1999, 2014), using the same approach adopted in this work.

A standard SELEN[4] output is shown in Figure 5, where the Earth's relief at the Last Glacial Maximum (LGM, $21,000$ years ago) is reconstructed in the post-processing phase of SELEN[4], according to the solution of the SLE. Note that the figure only shows the bedrock relief at the LGM, which is not what Peltier (1994) has called *true paleo-topography* (PT), which also includes the contribution of ice elevation. The user can easily obtain maps of the full PT merging maps like that shown in Figure 5 with those of Figure 1. At the global scale, the major land masses that were exposed at the LGM (Beringia,





Sunda, Sahul, and Doggerland) are clearly visible, as evidenced by the low elevation green areas. Also visible is the exposed continental shelf along the coasts of Patagonia, which has been the subject of investigation by Peltier and Drummond (2002) in the framework of GIA modeling. Light blue areas across the polar regions covered by thick ice at the LGM correspond to places where the ice was grounded below sea level at that epoch.

By increasing the spatial resolution, SELEN[4] can also be safely employed to resolve the past sea-level variations on regional scale. As a specific case study, we consider the Mediterranean Sea. The history of RSL in the Mediterranean Sea has been the subject of various investigations, stimulated by the amount of high-quality geological, geomorphological and archaeological indicators in the region (see *e.g.,* Lambeck and Purcell, 2005; Antonioli et al., 2009; Evelpidou et al., 2012; Vacchi et al., 2016; Roy and Peltier, 2018, and references therein). Since on the global scale of Figure 5 the details of the paleo-topography in this

area are difficult to visualize, we have used the outputs of the high-resolution run with settings *R100/L512/I5*, already exploited in Section 3.2.1. The results are shown in the map of Figure 6, where paleo-topography is shown at 26 ka. The vastly exposed continental shelf of Tunisia (Mauz et al., 2015) and the northern Adriatic Sea (Lambeck and Purcell, 2005) are now clearly visible, along with other smaller scale regions where the topography has seen significant changes during the last deglaciation (Lambeck, 2004; Purcell et al., 2007).

### 3.2.3   Polar motion

With SELEN[4], three configurations are possible, in which rotational effects on GIA are dealt with in different manners. First, these effects can be simply ignored, as it is done in the classical FC76 GIA theory. However, when rotational effects are taken into consideration, this can be done in two different ways, *i.e.,* either following the *traditional rotation theory* (Milne and Mitrovica, 1998; Spada et al., 2011) or a *revised rotation theory* proposed by Mitrovica et al. (2005) and Mitrovica and

Wahr (2011). The reader is referred to the literature for a detailed presentation of the two theories and to S5.2 for a brief account. Here it is useful to mention that in the traditional treatment the long-term response of the Earth is evaluated assuming that the lithosphere is characterized by a finite elastic strength, while in the revised theory the equilibrium rotational shape is, more realistically, only based on the viscous properties of the planet. Furthermore, the long term extra-flattening due to mantle dynamics is properly accounted for. We remark that in both cases the fast Chandler wobble component of polar motion

is filtered out since the onset from the Liouville equations, since the time scales of GIA largely exceed the Chandler wobble period ($\sim 14$ months, see *e.g.,* Lambeck 1980). The implications for the GIA response of the new theory are quite significant, as illustrated in detail by Mitrovica et al. (2005) and Mitrovica and Wahr (2011).

Solid curves in Figure 7 show the evolution of the polar motion components $(m_x, m_y)$ and their rates of change $(\dot{m}_x, \dot{m}_y)$ in response to GIA, obtained by solving the Liouville equations in the test run, in which the revised rotation theory is employed.

Further results, shown by dashed curves, have been obtained adopting the traditional theory. As the $x$ and $y$ components of polar motion are conventionally measured along the Greenwich meridian and $90°$E, respectively, Figure 7a indicates that since the inception of deglaciation, the displacement of the pole has been roughly in the direction of the Hudson Bay, consistent with the seminal results of Sabadini and Peltier (1981). The two theories predict similar evolutions of the pole of rotation, which according to Figure 7a has been displaced by $\sim 18$ km on the Earth surface by the glacial readjustment process since 26 ka.





We note that at the time of the rapid melting episode known as Melt Water Pulse MWP-1A (between 14.3 and 12.8 ka, see *e.g.,* Blanchon 2011) a sudden variation in the $m_y$ component of polar motion has occurred but no changes can be observed on $m_x$. When the *rates* of polar motion are considered in Figure 7b, differences between the predictions of the two rotations theories are more apparent. In particular, the present-day (0 ka) rate of polar motion are found to be $\sim 1$ and $\sim 3$ deg Ma$^{-1}$

for the revised and the traditional theories, respectively, which fits the predictions of Mitrovica and Wahr (2011) and confirms that the traditional theory largely overestimates the effects of GIA on polar motion. We note also that MWP-1A has caused a remarkable acceleration of polar motion, with a variation in the rate of $\sim 4$ deg Ma$^{-1}$ during a few centuries. As far as we know, the geophysical consequences of such sudden acceleration in the pole path have not been investigated yet.

### 3.3  Present-day GIA

In this section we describe further outputs of the SELEN$^4$ test run, focussing in the effects of GIA at present time. In particular, we shall consider *i)* the global pattern of the so-called *GIA fingerprints*, *ii)* predictions of the the rate of sea-level change at tide gauges, and *iii)* the time-variations of the Stokes coefficients of the Earth's gravity field induced by GIA.

#### 3.3.1  GIA fingerprints

Figure 8 shows another standard output of SELEN$^4$, *i.e.,* the present-day rates of variation of four fundamental quantities

associated with GIA, obtained for the test run. These are relative sea level ($\dot{\mathcal{S}}$, frame a), vertical displacement of the crust ($\dot{\mathcal{U}}$, b), absolute sea level ($\dot{\mathcal{N}}$, c) and the geoid height ($\dot{\mathcal{G}}$, d). After Plag and Jüettner (2001), these are often referred to as *GIA fingerprints*; their spatial variability reflects the effects of deformation, gravitational attraction, and rotation within the system composed by the solid Earth, the oceans and the ice sheets (Clark et al., 1978; Mitrovica and Milne, 2002). In view of their importance on the interpretation of ground-based (King et al., 2010) or satellite geodetic observations (Peltier, 2004) and

of tide-gauge secular trends (*e.g.,* Spada and Galassi, 2012; Wöppelmann and Marcos, 2016), their properties have been the subject of various investigations during last decade (see *e.g.,* Mitrovica et al., 2011; Tamisiea, 2011; Spada and Galassi, 2015; Spada, 2017; Husson et al., 2018; Melini and Spada, 2019).

It should be remarked that the four fingerprints shown Figure 8 are not independent of one another. In particular, the SLE gives $\dot{\mathcal{S}} = \dot{\mathcal{N}} - \dot{\mathcal{U}}$ according to Eq. (20). Furthermore, $\dot{\mathcal{N}} = \dot{\mathcal{G}} + \dot{c}$, where $c$ is the spatially invariant term introduced by FC76

to ensure mass conservation (see S2.4 in SSM19). The two relationships above hold regardless the particular combination of rheology and ice model employed, and the preferred rotation theory adopted. However, the patterns of the fingerprints and the numerical value of $\dot{c}$ are model-dependent. Other interesting results hold for the spatial averages of the fingerprints, which reflect some physical aspects of GIA (Spada, 2017) and are useful to correct geodetic observations from the effects of deglaciation (*e.g.,* Spada and Galassi, 2015). In Table 4 we summarise the numerical values of whole Earth surface averages

(denoted by symbol $< \cdots >^e$) and ocean-averages ($< \cdots >^o$) of the GIA fingerprints in the test run. In addition, we have also executed SELEN$^4$ adopting the traditional rotation theory and neglecting rotational effects, and the corresponding averages are shown in Table 4 as well. We note that by virtue of mass conservation $< \dot{\mathcal{G}} >^e = < \dot{\mathcal{U}} >^e = 0$ (see S4.3 and S6.2) regardless the rotation theory adopted, and as a consequence $< \dot{\mathcal{S}} >^e = < \dot{\mathcal{N}} >^e = \dot{c}$. We also note that the value of $< \dot{\mathcal{N}} >^o$, commonly





employed to correct the altimetry observations of absolute sea-level change for the effects of GIA, is in fair agreement with predictions from state-of-the-art GIA models (*e.g.,* Church et al., 2013a; Spada and Galassi, 2015; Spada, 2017). Notably, $< \dot{\mathcal{N}} >^o$ is not very affected by the choice of the rotation theory. Since model I6G-T05-R44 assumes that melting of the major ice sheets ceased $\sim 4,000$ years ago, the small value of $< \dot{\mathcal{S}} >^o$ only reflects on-going changes in the area of the oceans due

to GIA. In the FC76 fixed-shorelines approximation, $< \dot{\mathcal{S}} >^o$ would be identically zero by virtue of the mass conservation principle (see *e.g.,* Spada, 2017).

### 3.3.2    GIA at tide gauges

Estimating global mean sea-level rise in response to climate change requires the decontamination of tide-gauge records by the effects of GIA. Since the late $1980$s, with the awareness of global warming and the availability of numerical solutions of

the SLE (Peltier and Tushingham, 1989), GIA corrections to the observed trends of sea level have been routinely applied (for a review, see Spada and Galassi, 2012; Spada et al., 2015; Wöppelmann and Marcos, 2016). However, since GIA models are progressively improved to provide a better description of reality, corrections at tide gauges are not given once and for all (Kendall et al., 2006; Tamisiea, 2011; Melini and Spada, 2019). Furthermore, new constraints from past sea level or modern geodetic observations have permitted gradual refinements (either by formal inverse methods or simply by trial and error) of

the two basic ingredients of GIA modeling, *i.e.,* the Earth rheological profile and the history of deglaciation since the LGM. Uncertainties in modeling are significant (Melini and Spada, 2019), which constitutes an additional motivation to improve the approach to GIA.

In Table 5, we show SELEN[4] predictions for $\dot{\mathcal{S}}$ at a few tide gauges, for the test run and other possible configurations as well. Here we only show results for the 23 sites that have been considered by Douglas (1997) in his redetermination of

global sea-level rise, which obey specific criteria that make then suitable to represent the trend of secular sea-level rise. The sites chosen by Douglas (1997) are located in the periphery of the regions covered by thick ice sheets at the LGM, since GIA predictions at sites formerly beneath the ice sheets are expected to be more affected by uncertainties in GIA modeling. This has been recently confirmed by Melini and Spada (2019). The post-processing phase of  SELEN[4] can be configured to handle any properly formatted input dataset with coordinates of geodetic points of interest, where all the variables considered

in Figure 8 can be evaluated. This can be useful, for instance, to estimate the effects of GIA on vertical movements at specific GPS points (Serpelloni et al., 2013) (modules for the computation of horizontal displacements shall be included in future releases). Comparing column *d)* with *b)* and *c)* (all these runs are characterized by the intermediate resolution configuration *R44/L128/I3*), we note that rotational effects are important at tide gauges; however from *b)* and *c)* we also note that differences between the revised and the traditional rotation theory generally do not exceed the $0.1$ mm yr$^{-1}$ level at the tide gauges

considered here. Comparing outputs in *b)* with the high-resolution run in *a)*, we note maximum differences of $0.03$ mm yr$^{-1}$, which further confirms the reliability of the test run.

Comparing the high-resolution results in Table 5 (*R100/L512/I5*, column *a)*) with those obtained using the ICE-6G_C(VM5a) model in the original implementation of WR Peltier[2], reported in column *e)*, we note a fair agreement between the two model

---

[2]See *http://www.atmosp.physics.utoronto.ca/~peltier/data.php* - last accessed 06 June 2019.





predictions. The differences are generally close to $0.2$ mm yr$^{-1}$ or slightly larger, and the two sets of predictions are coherent. The values of $\dot{\mathcal{S}}$ averaged over the tide gauges differ, but they are both $\leq 0.1$ mm yr$^{-1}$. More significant differences in the $\dot{\mathcal{S}}$ values are however apparent when we compare model predictions for sites located in the polar regions beneath the former ice sheets, which are not considered in Table 5. At these locations, the expected $\dot{\mathcal{S}}$ values are of the order of several mm yr$^{-1}$, due

to the large isostatic disequilibrium still associated with ice unloading. For example, at the tide gauge of Stockholm, we obtain a rate of $-4.43$ mm yr$^{-1}$ for the high resolution run *R100/L512/I5* while in the original ICE-6G_C(VM5a) implementation, the rate is $-3.75$ mm yr$^{-1}$. We have also ascertained that misfits of the order of $1$ mm yr$^{-1}$ are not uncommon in other high-latitude sites of both hemispheres. Disclosing the origin of the discrepancies in the two sets of GIA predictions (and therefore in the whole set of the GIA geodetic fingerprints considered in Figure 8) is not easy at this stage, and would demand a detailed

model inter-comparison study like those performed in the GIA community by Spada et al. (2011) and Martinec et al. (2018). We can however guess that the misfit between the two sets of GIA predictions stems from the different discretisations of the ice time-histories, from the effects of mantle compressibility, and possibly from the different rotation theories adopted.

### 3.3.3   Stokes coefficients of the gravity field

In Figure 9 we study the present-day rates of change of the variations of Stokes coefficients $(\dot{\overline{\delta c}}_{lm}, \dot{\overline{\delta s}}_{lm})$ induced by GIA,

computed in the test run with *R44/L128/I3*. These quantities represent the coefficients of the expansion of the geopotential variation $\Phi(\gamma, t)$ in series of spherical harmonics, hence they contain information upon the response of the Earth to surface loading and to movements of the axis of rotation. In SELEN[4] we use a real, fully normalised representation for the Stokes coefficients, following the Gravity Recovery and Climate Experiment (GRACE) conventions for spherical harmonics (see, in particular, Bettadpur 2018 and S8.9). However, it is important to note that in Figure 9, the Stokes coefficients also include the

direct effect of Earth rotation on the degree 2 TLNs (*i.e.,* they account for the '$\delta(t)$' term in '$\delta(t) + k_2^T(t)$'), hence the rates we have computed are not directly comparable with the GRACE rates. In fact, since in its orbit GRACE is not physically connected with the Earth, it cannot be influenced by the direct rotational effect (the whole issue has been the subject of discussion a few years ago, see Chambers et al., 2010; Peltier et al., 2012; Chambers et al., 2012). The user of SELEN[4], however, can supply the program with a rotation response function $\mathcal{G}^{rot}$ that does not include the direct term in order to produce GRACE-compliant

Stokes coefficients which are only indirectly affected by Earth rotation.

The fully normalised cosine (squares) and sine (circles) coefficients $(\dot{\overline{\delta c}}_{lm}, \dot{\overline{\delta s}}_{lm})$ are shown in Figure 9a only for harmonics with degree $l \leq 6$. The dominance of the degree 2 coefficients is apparent, which reflect the symmetries of the $\dot{\mathcal{G}}$ fingerprint in Figure 8d. We note that since $\dot{\mathcal{N}} = \dot{\mathcal{G}} + \dot{c}$ where $\dot{\mathcal{N}}$ is the absolute sea-level fingerprint in Figure 8c and $c$ is the FC76 constant (see Eq. 21), the Stokes coefficients for $\dot{\mathcal{G}}$ and for $\dot{\mathcal{N}}$ coincide for $l \geq 2$. For reference, the numerical values of the degree $l = 2$

coefficients obtained in the test run are $\dot{c}_{20} = +1.59$, $\dot{c}_{21} = -0.76$, $\dot{s}_{21} = +3.37$, $\dot{c}_{22} = -0.35$, and $\dot{s}_{22} = +0.07$ in units of $10^{-11}$ yr$^{-1}$; the modulus of all other coefficients is $< 1$ in these units. To better study the decay of the Stokes coefficients with increasing $l$, in the diagram of Figure 9b we show the harmonic spectrum defined in S8.9. By inspection of the spectrum plot it is now apparent that the energy contained in the degree $l = 2$ harmonic component exceeds by at least one order of magnitude all those with $l \geq 3$. After a *plateau* that indicates a substantial power equipartition in the range of harmonics $3 \leq l \leq 7$, the





spectrum clearly shows a red character and decays very rapidly, closely following a power law $\sim l^{-5}$ (solid line). This result is consistent with those obtained by Spada and Galassi (2015), although they have used a simplified GIA model with fixed shorelines and the traditional rotation theory. They confirm that, for $l \geq 3$, the power contained in the GIA-induced regional variations in absolute sea level is negligible when compared with the spectrum observed during the altimetry era.

**4   Conclusions**

We have presented an updated version of the SLE solver `SELEN`, which has been originally introduced by Spada and Stocchi (2007) and principally meant as a tool for students. Along with a condensed theory background and the basic features of the new program, we have provided a step-by-step description of the outcomes of a medium-resolution *test run* that requires modest computing resources. However, the run accounts for an up-to-date description of the time history of melting since
the Last Glacial Maximum and a realistic rheological profile, being based upon a realisation of model ICE-6G_C (VM5a) of Peltier et al. (2015). The outputs of the test run, which cover different temporal scales, have been briefly discussed in order to appreciate some of the possible geodynamical implications. Outputs of a high-resolution test runs have been also presented to illustrate the effects of spatial and harmonic resolution on some GIA predictions.

With respect to the original version of the code, in SELEN[4] two major improvements have been made. The first is represented
by an increased physical realism in the description of the GIA process. Indeed, now the program accounts for the migration of the shorelines and for the rotational feedback on sea-level change, which enable a fully topographically and gravitationally self-consistent modelization of GIA (in the sense defined by Peltier, 1994). Furthermore, SELEN[4] can be configured assuming two different rotation theories, or even excluding rotational effects. The second improvement is in terms of usability, efficiency and versatility, and covers various aspects. First, the solution of the SLE is now performed by a single Fortran program unit,
leaving to a flexible and customizable post-processor the computation of various outputs encompassing the broad spectrum of the GIA phenomenology. Second, on modern multi-core systems, SELEN[4] can take advantage of multi-threaded parallelism to speed up the most computationally intensive portions of the code. Third, the user can easily customize the time-evolution of the surface load and the rheological layering of the Earth providing pre-computed loading and tidal Love numbers. Last, SELEN[4] comes with a User guide and with a fully detailed theory background, which is particularly meant to illustrate the
basic concepts of GIA to young scientists or colleagues and to allow transparency and reproducibility.

For simplified surface loads, recently a preliminary version of the new program has been successfully tested against other independently developed, but not yet publicly available, SLE solvers (see Martinec et al., 2018). After it has been progressively developed in various *interim* versions, SELEN[4] is now released to the GIA and to the global geodynamics community as an open source tool.

*Acknowledgements.* GS is funded by a FFABR (Finanziamento delle Attività Base di Ricerca) grant of MIUR (Ministero dell'Istruzione, dell'Università e della Ricerca) and by a research grant of DISPEA (Dipartimento di Scienze Pure e Applicate) of the Urbino University



"Carlo Bo". We thank D. Riposati from the INGV Laboratorio Grafica e Immagini for drawing the SELEN logo. We thank M. Tegmark for having made available his pixelization routines, which have had an essential role in the development of `SELEN` (see *https://space.mit.edu/ home/ tegmark/ isosahedron.html*). We also thank M. Wieczorek for distributing the SHTOOLS (the Spherical Harmonics Tools) to the community (*https://shtools.oca.eu/shtools*). Some of the figures have been drawn using the Generic Mapping Tools (GMT) of Wessel and

5   Smith (1998). We are indebted to all the colleagues who participated to the various stages of the Glacial Isostatic Adjustment and Sea Level Equation benchmark activities, namely: V. R. Barletta, P. Gasperini, T.S. James, M.A. King, S.B. Kachuck, V. Klemann, B. Lund, Z. Martinec, R.E.M. Riva, K. Simon, Y. Sun, L.L.A. Vermeersen, W. van der Wal and D.Wolf (see Spada et al., 2011; Martinec et al., 2018). A special acknowledgement goes to F. Colleoni for help in the code implementation and to F. Mainardi for advice on the theory of linear visco-elasticity. We are also indebted to M. Bevis and E. Ivins for encouragement and advice. R. Mascetti has patiently revised the manuscript

10  during various stages of its development, also providing invaluable inspiration.

*Code and data availability.* SELEN[4] is available from Zenodo at the link *https:// zenodo.org/ record/ 3339209* (DOI: 10.5281/ zenodo. 3339209) and from the Computational Infrastructure for Geodynamics (CIG) at *github.com/ geodynamics/selen*. The ice history data for ICE-6G (VM5a) have been downloaded from *http://www.atmosp.physics.utoronto.ca/ ~peltier/ data.php* (last accessed 20 Apr 2019). The Model ETOPO1 has been obtained from *https://www.ngdc.noaa.gov/ mgg/ global/* (last accessed 26 Feb 2019).

15  *Copyright statement.* SELEN[4] is released under a 3-Clause BSD License (for details, see *https:// opensource.org/ licenses/ BSD-3-Clause)*.

*Author contributions.* G.S. and D.M. have both contributed to the design and implementation of the research, to the analysis of the results and to the writing of the manuscript. The supplement has been written by G.S. with the support of D.M. The code has been progressively developed by G.S. and D.M., who has edited the User Guide and the on-line version of the code.

*Competing interests.* The authors declare no competing interests.



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





| *Test run* feature | Parameterisation used | Notes and links |
|---|---|---|
| Ice model | I6G-T05-R44 | Section 3.1.1 and Figure 1 |
| Rheology | VM5i | Section 3.1.3 and Table 2 |
| Compressibility | No | Spada et al. (2011) |
| Mantle layers | $N_v = 9$ | Table 2 |
| Density and shear modulus | PREM-averaged | Table 2 |
| Lithosphere / Core | Elastic / Fluid inviscid | Spada et al. (2011) |
| Reference frame origin | Center of mass (CM) | Spada et al. (2011) |
| Final topography | ETO-R44 | Section 3.1.2 and Figure 2 |
| Rotational effects | Revised theory | See S5.2 in SSM19 |
| Tegmark resolution | $R = 44$ | See S8.6 |
| Maximum harmonic degree | $l_{max} = 128$ | See S8.6 |
| External / internal iterations | $n_{ext} = n_{int} = 3$ | See S8.7 |

**Table 1.** Details of the configuration of the SELEN[4] *test run* whose results are considered in Section 3, with notes and links to text and figures.



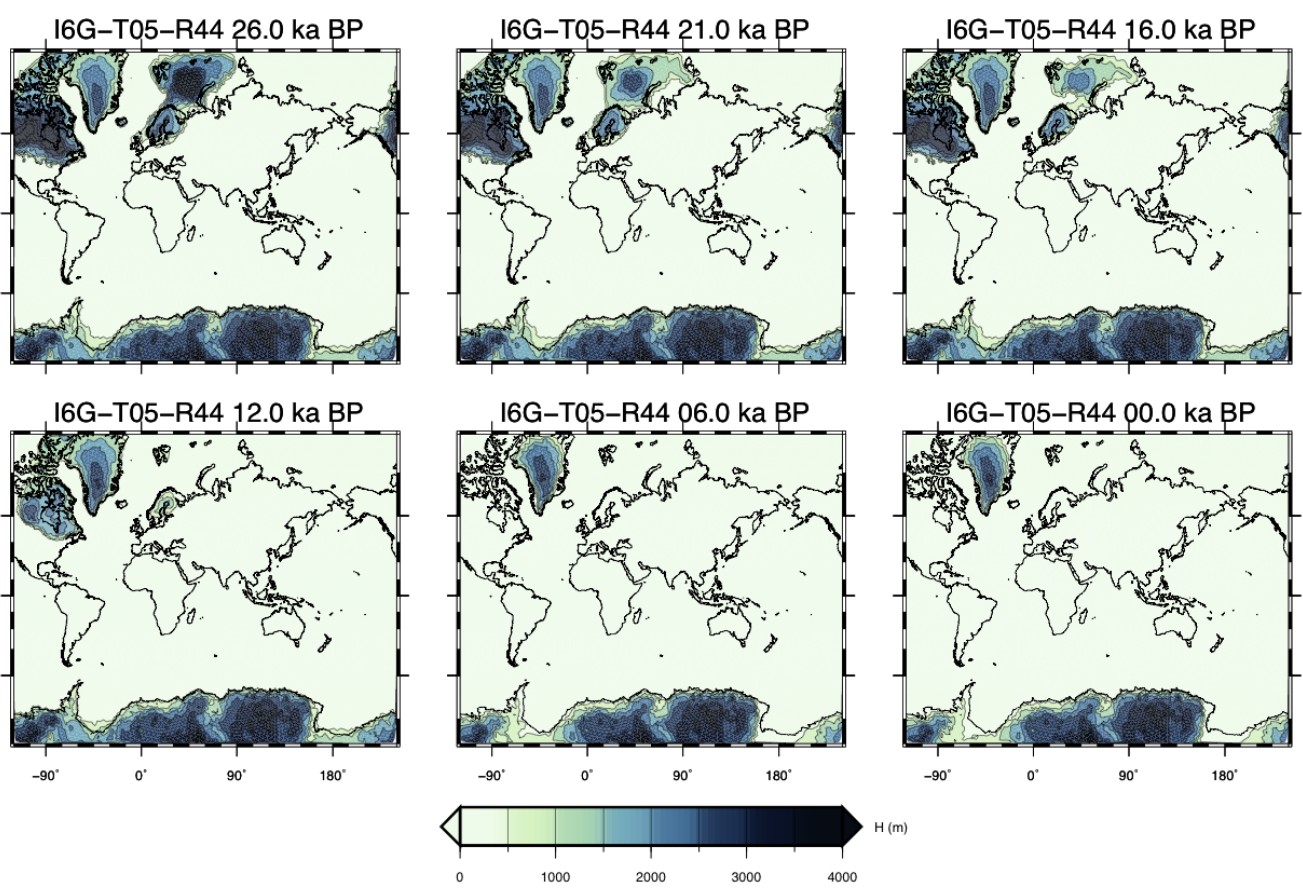

**Figure 1.** Ice distribution according to model I6G-T05-R44, at six different epochs since 26 ka. The maps are obtained by direct triangulation of the pixelized ice thickness data using the GMT program `pscontour`. This and the following figures are drawn using GMT scripts adapted from those which are available in the output folders of SELEN[4] after execution.



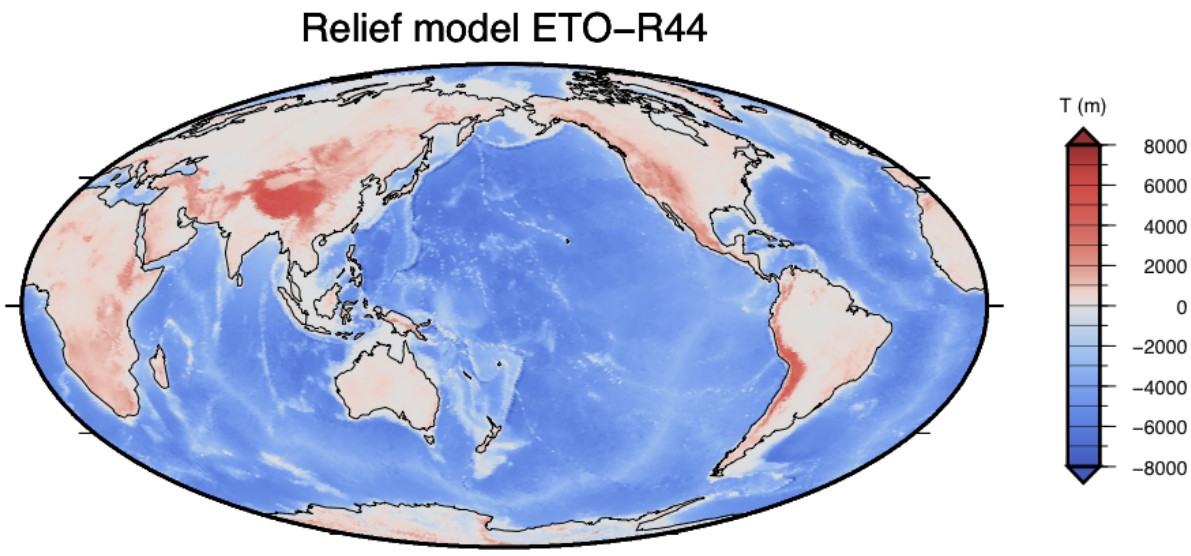

**Figure 2.** Present day relief according to model ETO-R44 used in the test run, obtained from model ETOPO1 by bilinear interpolation on the pixels of the Tegmark grid with resolution $R = 44$, using the GMT program `grdtrack`.





| Radius, $r$ | Density, $\rho$ | Rigidity, $\mu$ | Viscosity, $\eta$ | Layer |
|---|---|---|---|---|
| (km) | (kg m$^{-3}$) | (Pa $\times 10^{11}$) | (Pa $\cdot$ s $\times 10^{21}$) | |
| 6371.000 | 3192.800 | 0.596 | $\infty$ | LT |
| 6281.000 | 3369.058 | 0.667 | 0.5 | UM1 |
| 6151.000 | 3475.581 | 0.764 | 0.5 | UM2 |
| 5971.000 | 3857.754 | 10.647 | 0.5 | TZ1 |
| 5701.000 | 4446.251 | 17.027 | 1.5 | LM1 |
| 5401.000 | 4615.829 | 19.125 | 3.2 | LM2 |
| 5072.933 | 4813.845 | 21.242 | 3.2 | LM3 |
| 4716.800 | 4997.859 | 23.253 | 3.2 | LM4 |
| 4332.600 | 5202.004 | 25.540 | 3.2 | LM5 |
| 3920.333 | 5408.573 | 27.940 | 3.2 | LM6 |
| 3480.000 | 10931.731 | 0 | 0 | Core |

**Table 2.** Density, rigidity and viscosity profiles adopted in the rheological model VM5i, where abbreviations LT, UM, TZ and LM stand for lithosphere, upper mantle, transition zone and lower mantle, respectively. Some spectral properties of model VM5i, which constitutes a realisation of the original viscosity profile VM5a of Peltier et al. (2015), are given in Figure 3 and in Table 3.

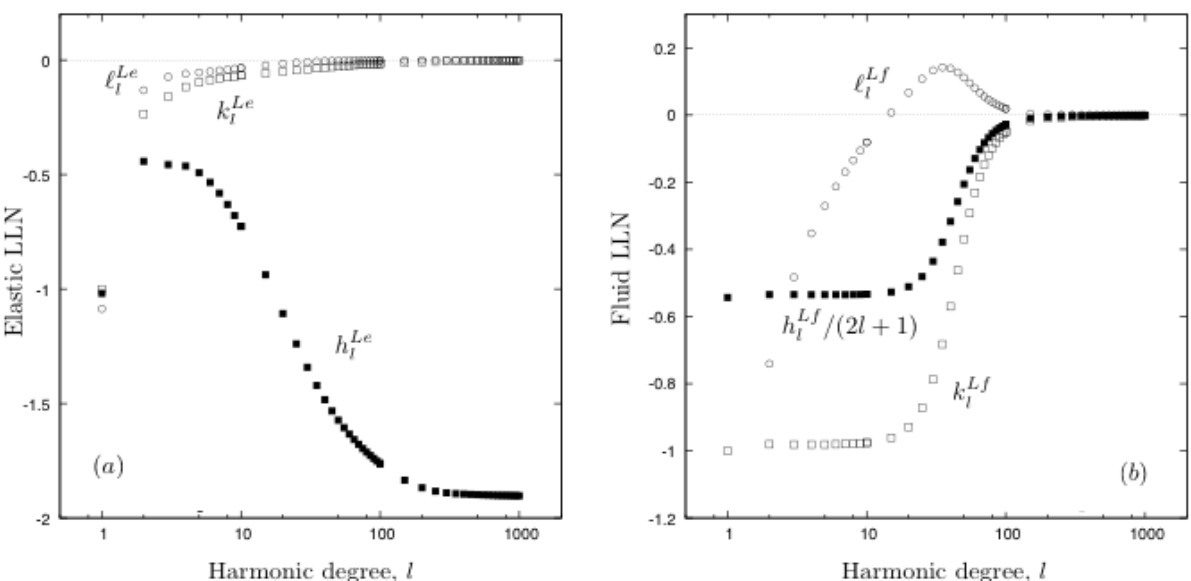

**Figure 3.** Elastic (a) and fluid (b) LLNs as a function of harmonic degree $l$ for the 11-layer rheological model VM5i employed in the test run (see Table 2). It is apparent that for this model asymptotic values are reached, in both cases, for $l$ exceeding a few hundreds. Note that in (b), where the fluid LLN for vertical displacement is normalised by $(2l+1)$, the relationship $h_l^{Lf} \approx (2l+1)k_l^{Lf}$ is apparent.



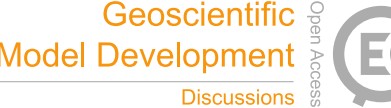

|  | $l=1$ | 2 | 4 | 16 | 64 | 128 | 256 | 512 | 1024 |
|---|---|---|---|---|---|---|---|---|---|
| $k_l^{Le}$ | $-1.000^0$ | $-0.235^0$ | $-0.116^0$ | $-0.557_1$ | $-0.227_1$ | $-0.123_1$ | $-0.641_2$ | $-0.323_2$ | $-0.162_2$ |
| $h_l^{Le}$ | $-0.101^1$ | $-0.442^0$ | $-0.463^0$ | $-0.974^0$ | $-0.165^1$ | $-0.180^1$ | $-0.188^1$ | $-0.189^1$ | $-0.190^1$ |
| $\ell_l^{Le}$ | $-0.108^1$ | $-0.131^0$ | $-0.600_1$ | $-0.207_1$ | $-0.189_2$ | $-0.451_3$ | $-0.579_4$ | $-0.108_4$ | $-0.272_5$ |
| $k_l^{Lf}$ | $-1.000^0$ | $-0.980^0$ | $-0.981^0$ | $-0.956^0$ | $-0.192^0$ | $-0.248_1$ | $-0.702_2$ | $-0.323_2$ | $-0.162_2$ |
| $h_l^{Lf}$ | $-0.163^1$ | $-0.267^1$ | $-0.481^1$ | $-0.173^2$ | $-0.139^2$ | $-0.363^1$ | $-0.198^1$ | $-0.189^1$ | $-0.190^1$ |
| $\ell_l^{Lf}$ | $-0.167^1$ | $-0.740^0$ | $-0.352^0$ | $+0.209_1$ | $+0.690_1$ | $+0.787_2$ | $+0.230_3$ | $-0.104_4$ | $-0.272_5$ |

**Table 3.** Numerical values of the LLNs for the rheological model VM5i (see Table 2), for some harmonic degrees $l$. We use the compact notation $v_e = a \times 10^{-v}$ and $v^e = v \times 10^e$, where $v$ is any value in the table and $e$ is an exponent. Note that, for this model, the elastic TLNs of degree $l = 2$ are $(k_2^{Te}, h_2^{Te}, l_2^{Te}) = (0.289^0, 0.524^0, 0.108^0)$ while the fluid values are $(k_2^{Tf}, h_2^{Tf}, l_2^{Tf}) = (0.931^0, 0.191^1, 0.514^0)$. The elastic residue of the Polar Motion Transfer Function (PMTF) is $A^e = 1.436$ while the secular residue is $A^s = 0$ since in the test run we employ the *revised rotation theory* for GIA (Mitrovica et al., 2005; Mitrovica and Wahr, 2011).



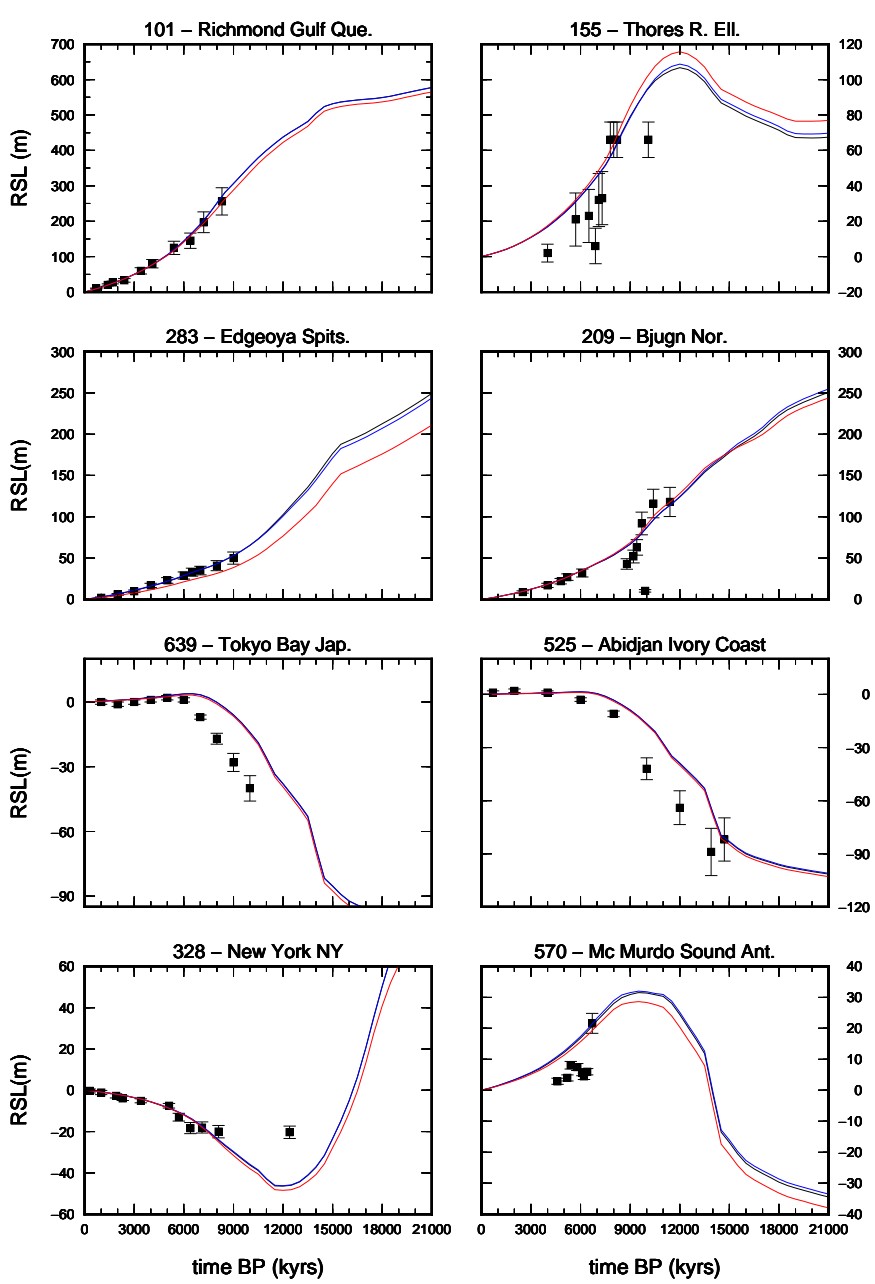

**Figure 4.** RSL data (with error bars) at eight of the 392 sites of the TP93 RSL database. Black curves show the results of the standard test run with configuration *R44/L128/I3*, the blue ones are for the high-resolution run with *R100/L512/I5* while those in red are for a low-resolution configuration with *R30/L64/I2*.

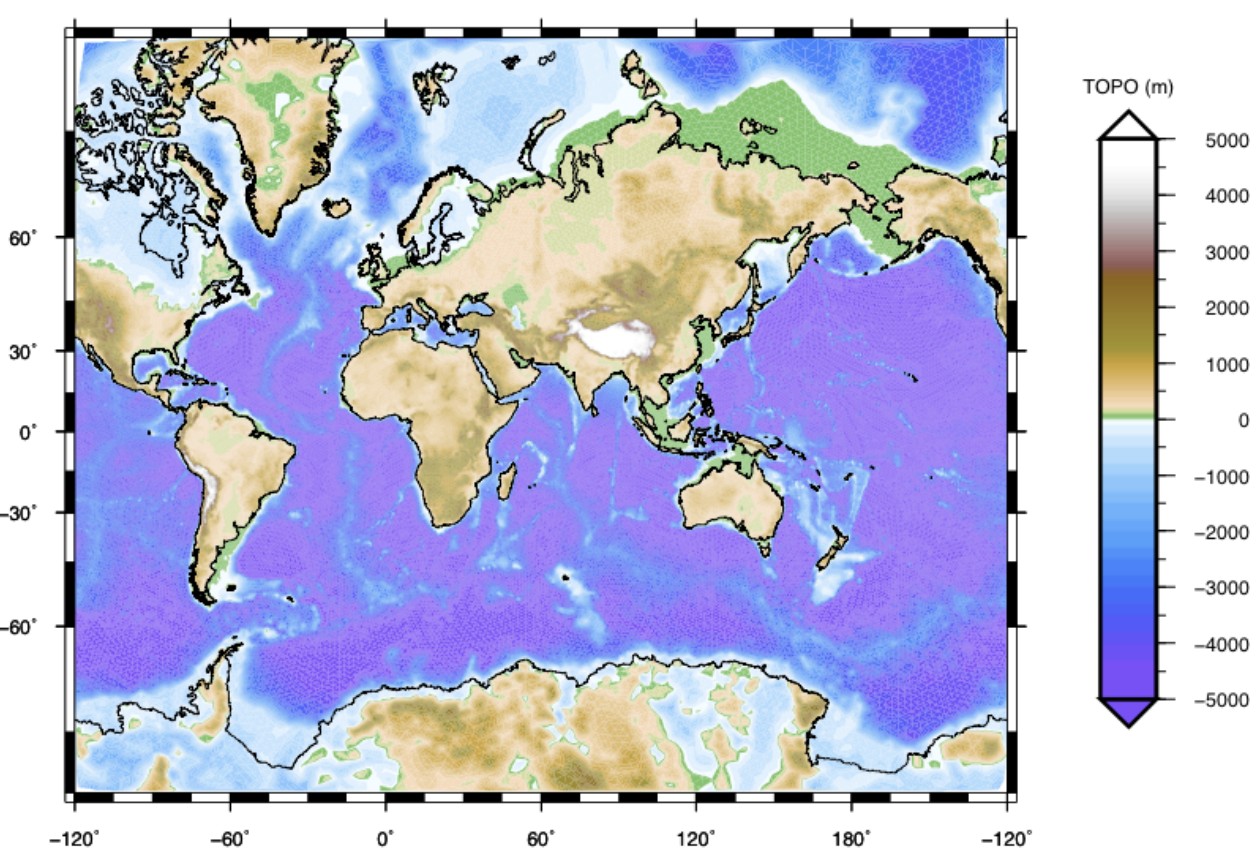

**Figure 5.** Sample SELEN[4] output of the global relief at the LGM (21 ka), according to the test run with *R44/L128/I3*, where topography ETO-R44 of Figure 2 has been used as a final condition for the SLE.



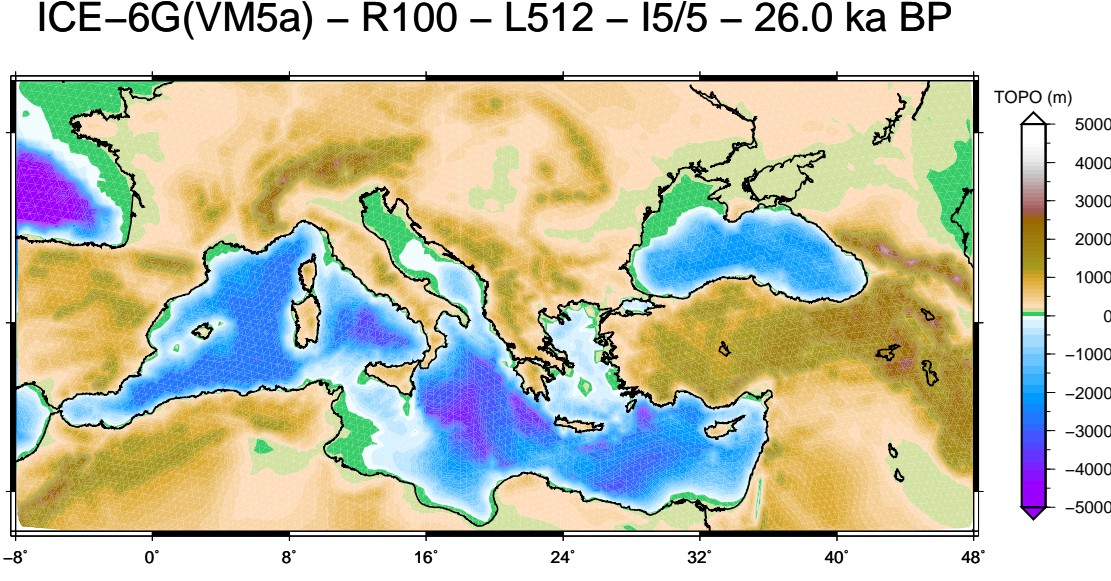

**Figure 6.** Paleo-topography of the Mediterranean Sea and of the Black Sea at 26 ka, obtained by a high-resolution SELEN[4] run with configuration *R100/L512/I5*, in order to enlighten the exposed lands in detail.





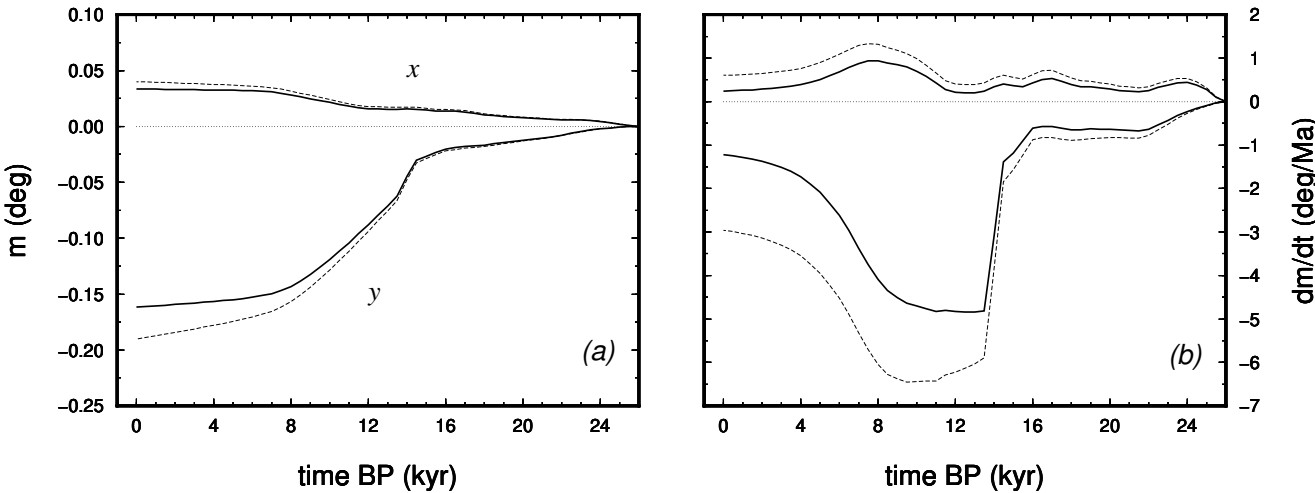

**Figure 7.** Cartesian components of polar motion along the axes $x$ and $y$ as a function of time *(a)* and their time derivatives *(b)* since the beginning of deglaciation, for the test run with configuration *R44/L128/I3*. Dashed and solid curves show results for the *traditional* and for the *revised* rotational theories, respectively. The steep change in the $y$ components at $\sim 14$ ka is forced by the inertia variations due to the occurrence of the Melt Water Pulse MWP-1A.



**Figure 8.** Present-day GIA fingerprints obtained for the test run with *R44/L128/I3*. Note that the color table is saturated in a narrow interval. The effects of Earth rotation, evaluated according to the revised rotation theory, can be well discerned for $\dot{\mathcal{N}}$ and $\dot{\mathcal{G}}$, with the characteristic high-amplitude lobes with a harmonic degree $l = 2$ and order $m = 1$ symmetry (Spada and Galassi, 2015). Spatial averages of these maps are given in Table 4.





| Average | a) New theory | b) Traditional theory | c) No rotation |
|---|---|---|---|
| | (mm yr$^{-1}$) | (mm yr$^{-1}$) | (mm yr$^{-1}$) |
| $<\dot{\mathcal{S}}>^e$ | $-0.20$ | $-0.20$ | $-0.19$ |
| $<\dot{\mathcal{U}}>^e$ | $+0.00$ | $+0.00$ | $+0.00$ |
| $<\dot{\mathcal{N}}>^e$ | $-0.20$ | $-0.20$ | $-0.19$ |
| $<\dot{\mathcal{G}}>^e$ | $+0.00$ | $+0.00$ | $+0.00$ |
| $<\dot{\mathcal{S}}>^o$ | $-0.01$ | $-0.01$ | $-0.01$ |
| $<\dot{\mathcal{U}}>^o$ | $-0.24$ | $-0.27$ | $-0.22$ |
| $<\dot{\mathcal{N}}>^o$ | $-0.25$ | $-0.27$ | $-0.23$ |
| $<\dot{\mathcal{G}}>^o$ | $-0.05$ | $-0.08$ | $-0.03$ |

**Table 4.** Whole Earth and ocean averages of the GIA fingerprints according to the test run based upon the new rotation theory (column *a)*), and the traditional theory *b)*. In *c)* we also consider the case when no rotational effects are taken into account. In all the computations we have adopted the combination *R44/L128/I3*. Note that $<\dot{\mathcal{N}}>^e = \dot{c}$, where $c$ is the FC76 constant. In this table, the SELEN[4] outputs are rounded to two significant figures.





| Tide gauge site | *a) R100/L512/I5*<br>Revised theory<br>$\dot{S}$ (mm yr$^{-1}$) | *b) R44/L128/I3*<br>Revised theory<br>$\dot{S}$ (mm yr$^{-1}$) | *c) R44/L128/I3*<br>Traditional theory<br>$\dot{S}$ (mm yr$^{-1}$) | *d) R44/L128/I3*<br>No rotation<br>$\dot{S}$ (mm yr$^{-1}$) | *e) ICE-5G_C(VM5a)*<br>$l_{max} = 512$<br>$\dot{S}$ (mm yr$^{-1}$) |
|---|---|---|---|---|---|
| 1. Newlyn | +0.19 | +0.18 | +0.17 | +0.13 | +0.37 |
| 2. Brest | +0.26 | +0.26 | +0.25 | +0.21 | +0.26 |
| 3. Cascais | +0.01 | +0.01 | +0.00 | −0.05 | −0.05 |
| 4. Lagos | −0.14 | −0.14 | −0.14 | −0.14 | +0.02 |
| 5. S.C. Tenerife | +0.16 | +0.14 | +0.13 | +0.06 | +0.15 |
| 6. Marseille | +0.09 | +0.07 | +0.06 | +0.05 | +0.05 |
| 7. Genova | −0.01 | +0.00 | +0.00 | −0.01 | −0.07 |
| 8. Trieste | −0.11 | −0.11 | −0.11 | −0.10 | −0.15 |
| 9. Auckland | −0.18 | −0.19 | −0.20 | −0.24 | −0.35 |
| 10. Dunedin | −0.18 | −0.19 | −0.21 | −0.26 | −0.40 |
| 11. Wellington | −0.21 | −0.24 | −0.25 | −0.29 | −0.36 |
| 12. Honolulu | −0.07 | −0.07 | −0.07 | −0.09 | −0.09 |
| 13. San Francisco | +0.62 | +0.60 | +0.58 | +0.48 | +0.31 |
| 14. Santa Monica | +0.47 | +0.46 | +0.44 | +0.33 | +0.20 |
| 15. La Jolla | +0.43 | +0.43 | +0.40 | +0.30 | +0.20 |
| 16. San Diego | +0.43 | +0.43 | +0.40 | +0.29 | +0.21 |
| 17. Balboa | −0.07 | −0.09 | −0.10 | −0.15 | −0.11 |
| 18. Cristobal | −0.06 | −0.07 | −0.08 | −0.13 | −0.07 |
| 19. Quequen | −0.42 | −0.42 | −0.39 | −0.22 | −0.51 |
| 20. Buenos Aires | −0.52 | −0.52 | −0.49 | −0.35 | −0.49 |
| 21. Pensacola | +0.63 | +0.63 | +0.60 | +0.46 | +0.49 |
| 22. Key West | +0.27 | +0.27 | +0.24 | +0.12 | +0.25 |
| 23. Fernandina | +0.59 | +0.58 | +0.55 | +0.41 | +0.40 |
| Average | +0.09 | +0.09 | +0.08 | +0.04 | +0.01 |

**Table 5.** Present-day rates of sea-level change at the 23 Douglas (1997) tide gauges, for the test run of SELEN[4] (column *b)*) and for other configurations. Results based upon the original implementation of model *ICE-5G_C(VM5a)* are reproduced in column *e)*. The average rate is also shown in the bottom line. The SELEN[4] outputs have been rounded to two significant figures.



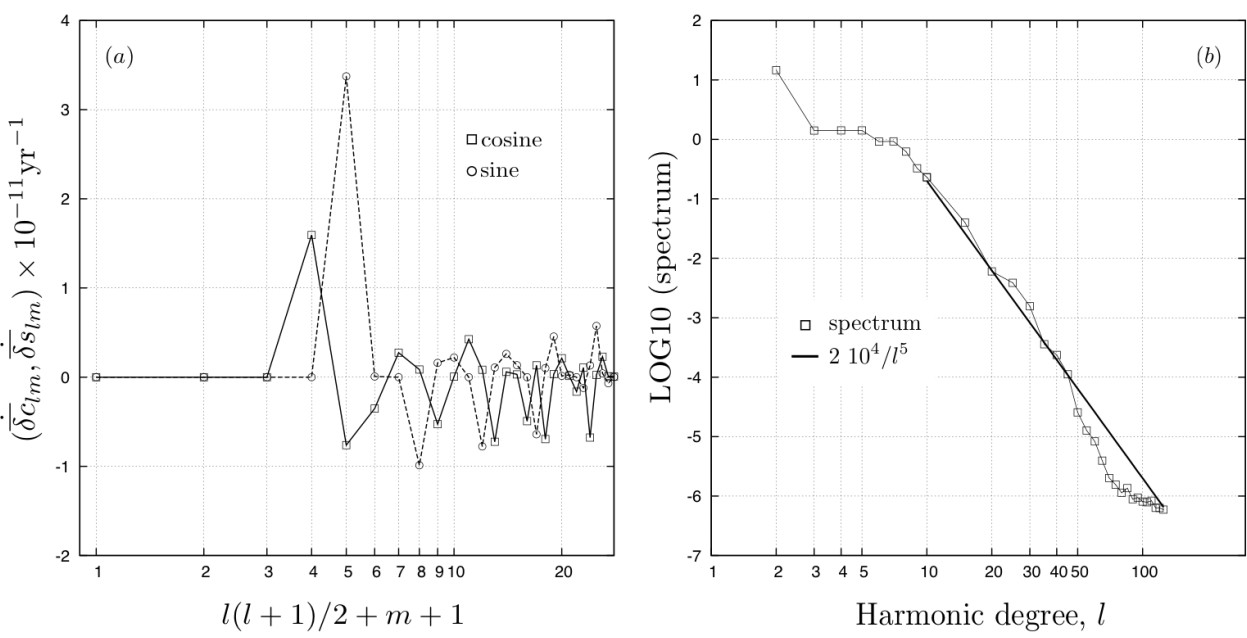

**Figure 9.** Low-degree ($l \leq 6$) Stokes coefficients obtained for the test run of SELEN$^4$ (frame a) and their full spectrum extended to harmonic degree $l_{max} = 128$ (b). The solid line in (b) represents the power law that best-fits the spectrum (in the least squares sense), obtained for $l \geq 10$.