# Peer review of "SELEN4 (SELEN version 4.0): a Fortran program for solving the gravitationally and topographically self-consistent Sea Level Equation in Glacial Isostatic Adjustment modeling"

_Geoscientific Model Development, 2019_

## Referee Comment (RC1) · Volker Klemann (Referee) · 19 Aug 2019

General comments

The authors present in this manuscript an update of numerical code which enables to calculate the gravitationally consistent interaction between a surface mass load and water mass load which compensates the total mass change. Previous versions of this code were presented already on a number of workshops dealing with glacial isostatic adjustment, GIA. In contrast to these versions, the authors improved the code in a

number of aspects which are currently discussed in GIA, and extended the portability of the code a lot. These improvements, to my point of view justify a new publication in a method and code-oriented Journal like GMD.

As I don't understand this review as a testing of the code, I will focus my review purely on the presented manuscript, and will not consider the supplement in this regard, especially for the derivation. In general, the manuscript is clearly written, the focus lies on the considered theory which the authors present in the theory section, and, as an application, the authors chose a published and established forcing for which they discuss the output in detail. Due to the fact, that the manuscript presents a methodologically oriented study, they do not discuss deviations from their results to those presented in the original publication of this forcing. From my point of view, this is reasonable strategy. As a validation of the method they refer to the benchmark study Martinec et al., 2018, which was recently published, and to which they contributed with a preliminary version of this code version.

Of course the question may arise, why the authors did not present the results for the benchmark, and discuss the addition of the rotational effect which was not considered in that benchmark this would be a nice extension to that study. If they won't do this, at least they should state, that the results of that version are identical with the current one, if rotation is switched off. Consulting that study, the SELEN code deviates a bit more than the order models when considering moving coastlines and floating ice. But this is only a suggestion, to establish more a benchmark study, than the quite complicated ICE6G vm5a applied here.

In the following, I focus on individual aspects of the presentation.

to Introduction

P. 2, L 16: Elasticity is material law, but not a rheology.

to Theory

To present a reduced version of a derivation is every time dangerous, especially if this is not put into an appendix but a supplement which is not part of the manuscript. So, a number of questions arose, which partly come up in the following comments.

P. 4, L 7: You refer to SSM19, here you should at least specify on which principles this expression (Eq. 2) is based.

P. 4, L 10: The meaning of OF should be defined explicitly.

P. 4, L 12: You introduce here the term bedrock topography, but do not specify what it means, especially as you further down use this quantity, to derive changes in sea level.

P. 4, L 16: Here and in the following, you use the 'cal' symbol to specify variations with respect to a reference state. If so, you can of course reduce the number of equations, e.g., Eq.s 6, 11, 13 and 15 become redundant. Furthermore, you do not speficy the reference state itself.

P. 4, L 24: I don't htink that you have to refere to SSM19 to introduce the definition (7), but simply if follows from $\cal M / \int_e dA$.

P. 5, L 2: This is somehow abrupt, and do you really need it here?

P. 5, L 4: What do you understand as a plausible surface load. At least you should guide the reader a bit more than referring to Bevis et al. 2016.

P. 5, L13: Is 'stem' used here synonymously to 'is associated'? If so, why then using a different word.

P. 5, L 5ff: How do you ensure in (Eq. 9) that you considered all terms with respect to $\cal M$? In Eq. 5, I ike that \cal O can be one of \{-1, 0, 1\}.

P. 6, L 20ff: The definition of seal level from topography is a bit tricky to understand, as you did not define topography itself properly, also sea surface is not defined her properly. So, why not motivate it from Gauss definition of the geoid as that equipotential surface to which a static ocean surface would adjust? Then, the jump from N to r^ss

is not as large. Also at the beginning you should specify what the sea level equation expresses. Only then you can obtain its most basic form as U-N. Also the relation to the water column might help the reader to understand its relation to loading more intuitively.

P. 6, L 12: This is also known as Bruhns formula. As $\cal .$ is not applied to $Phi$, I suggest to write this equation a bit different: \cal G = \frac{\Phi - \Phi}{g_0}, to remain in the used schema of formula symbols.

P. 6, L 13: One important reference in your discussion of the SLE is missing which is Martinec and Hagedoorn, 2014, who discuss in detail the SLE of a rotating body. They, for instance, distinguish between gravity and gravitational potential, where the latter only considers the surface loading, i.e. variation due to surface mass change and solid earth deformation. The gravity potential, MH14 use for gravitational potential plus rotational effects. This is in accordance to Helmut Moritz, 1990, The figure of the earth. So I would use gravity potential instead of geopotential.

P. 6, L 14: The meaning of c only depending on $t$, can easily be explained simply by the fact, that the displacement of the reference potential surface, N_0, does not represent the potential level to which the water surface will adjust. This is not only due to mass conservation but also due to the changes in the ocean basin. So, it is a bit more complicate.

P. 7, L 10: Here and in the following, $O$ is dangerous to use, as it is easily mistaken for the reference state 0.

P. 7, L 12: From type setting it is nicer to use text abbrevations in formulas with an unslanted font, e.g., ${\cal S}^\text{ave}$.

P. 7, 14: For the definition of the different sea-levels see below. For me, equivalent is more a renormed quantity, but here you are using it with respect to a model dependent quantity, depending on the current ocean area A^o (\gamma, t) and \mu (gamma, t).

Both terms depend on the $\cal S$ and $\cal L^{[abc]}$.

P. 7, L 8ff: This derivation is quite interesting, and I have needed some time to understand this alternative expression, especially why in Eq. 33 $T\_0$ appears instead of $T$. May be, a specification what 'using for $c$ the expression found in SSM19' does mean, might help, as c does not appear anymore. The interesting aspect is that this form is independent of the syphoning, which dominates <S>ˆe, and is put into <G-U>ˆO. May be this might help to understand Table 4 a bit more. So I guess the expression, $- <G-U>ˆO + Sˆave = c$ is missing.

P. 8 L 3: Wikipedia defines the 'eustatic change' as the alteration of sea level due to changes in volume of water or ice, and also due to changes in ocean topography, which goes back to Suess. The article to my point of view is based also on Rovere et al. (2016, https://doi.org/10.1007/s40641-016-0045-7), that is, also steric changes are considered. I would call Eq. 36, the equivalent sea level, as you also do at P. 10, L. 14ff, expressing an ocean equivalent of the ice, which is simply a renormation of water mass expressed as water height. Sˆave, I like. Your Sˆeus, I would call \bar{\cal S}ˆice, what is the equivalent of ice at current state. Your Sˆequ is a help quantity, I would keep its definition. FC76 phrase Eq. (36) to be the eustatic sea level although the ocean basin might slightly change. The authors are aware of the fact and "tried to avoid [. . .] the word (eustasy) has received so many qualifications since it was introduced by Suess". So, I would suggest not to use eustasy at all. It is a deprecated definition of sea level and mainly leads to misunderstanding.

P. 8, L 17: 'essentially' is a bit enigmatic, do you mean terms of d/o = 2/0 and 0/0 can be neglected as they are too small? See also MH2014.

P. 8. L 7: From here to the end of this section, you discuss the numerical implementation into SELEN. As the whole article is about SELEN, I would expect a bit more detail, how the equations are solved also with respect to the iterations you discuss in the following. So, this part I suggest to extend.

to A test run of SELEN

In the introduction to this section you should mention that you also consider changes in rotation in addition to N, U and S.

P. 10, L. 23: Why do you use an interpolation. To go from a fine to a coarse grid by interpolation is everytime a bit dangerous. I would expect here a filtering or a binning algorithm.

P. 11, L 10: 'Not agreed results on ve. Love numbers'. Do you have a reference for this statement, and why this is an argument to neglect compressibility?

P. 11, L 15: I would simply start the sentence: 'Numerical values of [. . .] in Table 3 and its caption, respectively', to avoid using two times reference, and the reader to search for the tidal love numbers in the table rows.

P. 11, L 17 and P. 14, L 9: For the captions use 'Glacial isostatic adjustment in the past' and '[. . .] at present day' or alternatively 'Paleo glacial isostatic adjustment' and 'Present-day [. . .]'.

P 11, P 27: I would in one sentence introduce the three configurations you discuss in the following.

P. 13, L 2: Why do you mention the small region around Patagonia with a reference only. Why not citing Klemann et al. (2015, DOI 10.1007/s41063-015-0004-x) for the East Siberian shelf or Lambeck for the Sunda Shelf. Furthermore, the light blue areas may represent locations covered with grounded ice, but also floating ice might be present there. The ice extent is not shown in Fig. 5.

P. 13, L 20. As stated before the MH14 paper, is in my point of view an important contribution to the discussion of rotation in GIA.

P. 13, L 25: 'since' used twice.

P. 14, L 8: For such statement, you should at least relate the order to that of other

transient processes observed or proposed for rotational variations.

P. 14, L 11: Why 'shall', although you consider it.

P. 14, L 15: Here you define $\cal S$ as relative sea level, I suggest you to use this definition from the beginning, what also would clearify what the SLE is solving for.

P. 14, L 16: would repeat 'frame' also before b, c and d.

P. 15, L 4: This is only clear from the fact that the syphoning effect is inside the definition of $\gamma$-depending part, otherwise the reader would wonder.

P. 15, L 8: Instead of 'decontamination', geodesists speak about 'correction for' an effect.

P 15, L 20: 'then' to 'them'.

P. 16, L 23–25: This is an interesting aspect. Can you give a formula to indicate which term has to neglected as direct rotational effect?

P. 16, L 32: Is this the degree variance, and why not presenting the equation here?

to Conclusions

P. 16, L 23–25: This is an interesting aspect. Can you give a formula to indicate which term has to neglected as direct rotational effect?

P. 16, L 32: Is this the degree variance, and why not presenting the equation here?

P. 17, L 12: replace 'runs' by 'run'.

P. 17, L 15: Not sure, if one can speak about 'physical realism'.

to figures and tables

Figure 1: In this color scheme, the dark ice thicknesses make it difficult to identify the coast lines. In this caption and in the further ones, I would not state which plotting command you used.

Figure 2: I would skip this figure.

Table 2: 'Density, [. . .] values specifying the parameters of the homogeneous layers defining the adopted [. . .]'. 'Some spectral [. . .]' I would skip this sentence, as it does not contribute to the table content.

Table 3: The PMTF is not defined in the manuscript. Without it, the Aˆ[es] and [hkl]ˆT are not necessary. In this context it would be interesting, if you need also viscoelastic tidal love numbers.

Figure 8: Can you specify the extreme values reached in the respective plots?

Table 4: I would skip the integrals over the whole earth, as these are clear from the definition of U and G.

---

## Short Comment (SC1) · 21 Aug 2019

Page 6, line 6: There is a typo in equation 19 that defines the vertical displacement of the solid surface of the Earth: Index "ss" should be "se" referring to Earth's solid surface (se) instead sea surface (ss).

---

## Author Comment (AC1) · 21 Aug 2019

We thanks Dr. Bagge for this comment. Yes, it is a typo that shall be corrected in the revised version of the manuscript. Regards GS

---

## Short Comment (SC2) · 28 Aug 2019

I have been using SELEN for a couple of years now, and I want to thank Dr. Spada et al for making this useful SLE code open source. As mentioned in the abstract, this is the only open source SLE program (that I am aware of), which is surprising considering that other dynamic parts of the Earth system (i.e. ice sheet models, ocean models, atmosphere models) have many open source options. I hope that with the release of SELEN4, it will encourage other GIA modelling groups to follow suit.

[Figure]

The new version of SELEN has several improvements that bring it in line with other 1D Earth structure GIA models, namely migrating shorelines and rotational effects. Migrating shorelines is probably the most essential new feature, as migration of the shoreline affects the total ocean volume for calculating sea level, and also will impact the GIA signal on areas with extensive continental shelves.

I've tested out SELEN4, and found it relatively trivial to get running. The included documentation is clear and helpful with the installation issues I initially had. SELEN 2.9 was really easy to get working and modify for my purposes, but there were a lot of post-processing elements that were included directly in the code that led to unnecessary overhead. I am happy to see that these post-processing options in SELEN4 are now separated from the main GIA calculation. I think the example post-processing examples can be easily modified for the general user who wants to include some modelling results in their studies.

Perhaps the biggest problem a user might face is if they want to add in their own ice load. Publicly available models (ICE-3G, ICE-5G and ICE-6G) are included in the distribution, but there is no tool to create your own. I have written code to create the pixel files from an ice load in an arbitrary projected coordinated system (https://github.com/evangowan/selen/tree/master/pixel), but I have not yet tested if it is compatible with SELEN 4. If the authors have code that can make the ice load input files, I encourage them to make it available.

Best Regards,

Evan J. Gowan

---

## Author Comment (AC2) · 30 Aug 2019

We thank Dr. Gowan for the positive feedback about our work. In SELEN4, any ice sheet model can be implemented, provided that it is discretized into disc-shaped elements and given in the proper SELEN-compliant format, described in the User guide. For the moment, we have included a realization of the chronology of ICE-6G. Following Dr. Gowan's suggestion, we shall also provide other models of the ICE-X family in forthcoming releases. We are also planning, in a future version, to include a routine for the computation of the isostatic and rotational Green's functions. This shall be done

as soon as a manageable and easy-to-use code is available. We would be happy to test any ice model that Dr. Gowan ha produced, in order to verify its compatibility with SELEN4.

---

## Referee Comment (RC2) · Samuel Kachuck (Referee) · 5 Sep 2019

This manuscript details the background and significant improvements to an opensource implementation of the sea-level equation. As solutions to this equation are essential to understanding relative sea level and geodetic evolution, the open-source method and documentation are welcome and the changes - including the physics of rotation and shoreline migration and the modularization of the code - warrant publication.

In addition, I believe the inclusion of the derivation to be useful to the community. Pursuant to that, there are areas where the derivation could be made even clearer, as noted below. Occasionally, a proliferation of variable definitions and abbreviations clouds the exposition, and a simplification would be welcome.

Finally, as the authors have just used this code's predecessor to participate in the benchmarking exercises of Martinec, et al. 2018, I think updating the contributed results with the new capabilities is warranted. In particular benchmark E, in which it seems they did not participate. As this is perhaps a more technical detail, I think it should be satisfactory to put this in the supplement, with perhaps a comment in the introduction to section 3. To this end, it would also be very interesting to see some discussion of how the included physics of rotation affect the results of those benchmarking cases.

All further comments are presented in page-line format

Specific Comments

1-15: I would like to mention that the python module, giapy, which was benchmarked in the Martinec, et al. 2018 paper (including shoreline migration and iteratively determined initial topography, though without rotation) is available for download, opensource (from https://github.com/skachuck/giapy), and similarly uses any normal-mode form love numbers. It is safe to say, however, that it has not yet been as comprehensively documented in its open-source release as SELEN.

2-15: "viscoelastic rheology...properly taken into account" see note to 1-15

4-12: Could you use \$S\$ instead of \$B\$ to keep it all consistent?

4-2: "ice and by the water in the ocean" refers to ice on the continent and water in the ocean, which is the basis for equation (2). Equation (1) refers to ice and water over the surface of the earth, i.e., \$\_e\$.

4-13: given the detail of the surrounding derivation, could you spare a few words giving

a precise definition of "topography"?

5-4: The meaning of "plausible" is not addressed in the referred paper beyond setting the integrated mass load variation, or equivalently the zero-degree coefficient, to zero.

5-6: Could more interpretable labels be given to the components of the load variation than a, b, and c?

5-10: The repeated definition of the variation variables gets confusing. Could the font used throughout be defined as the variation with respect to some reference level?

5-19: Can you give a physical meaning to the height variable  $Q^?$  "Auxialliary variable" and "arbitrary reference density" are a little confusing and undermotivated. If you selected  $\gamma r=\gamma v^{+}, hen it takes the form of a free water column, positive where there is liquid water, negative where the water is frozen, and zero otherwise.$

7-9: Throughout, when referring to equations in the supplement, could you refer to specific equations?

9-12: Could you briefly describe the significance of the external and internal iterations?

16-12: You could cite somewhere the work done by Barletta and Bordoni (2013) on the effects of implementation of ice histories.

17-27: see note to 1-15.

Table 2: For clarity, you might articulate that these are piecewise constant layers whose upper radii are given by the variable \$r\$.

Figure 4: Is the ordering significant? If so, state what guides it. The non-numerical ordering makes it difficult to locate them from the text quickly.

Figures 5 and 6: the color scale looks different between these figures for the resolutions in the manuscript draft. Make sure to check this as it goes through editorial.

Technical corrections

СЗ

- 2-2: "available since" is a bit awkward, consider "dating from"
- 2-10: "Despite GIA is now tightly" missing clause between "Despite" and "GIA"
- 2-16: "limited the" missing word "to"
- 2-16: "elastic rheology" to "elastic deformation"
- 2-20: "Love numbers" to "Love number"
- 2-28: "is hosting" to "has hosted" for tense agreement
- 2-30: "Since year" missing work "the"
- 3-24: "taken by the SLE" to "the SLE takes"
- 3-24: remove "an"
- 6-6: Should \$^{ss}\$ be \$^{se}?

8-25: "accomplished projecting" missing word "by"

11-14: "with origin in the whole Earth's..." to "with the origin at the Earth's..."

12-7: "from high-resolution but also from" to "from both high-resolution and"

12-21: The first clause "Differently...program" is awkward to read and the sentence would be fine without it.

13-5: The meaning of the word "safely" is not clear.

13-25: "since" used twice in one sentence.

13-27: Consider adding "and shown in Figure 7" at the end of that sentence for a smoother transition.

13-30: The sentence beginning with "Further results" could be revised to "Dashed curves show results obtained with the traditional theory."

14-9: Inconsistent abbreviations in section headings; "GIA versus "Glaical Isostatic Adjustment"

14-25, 14-31: "regardless the" missing word "of"

15-3: "very affected by" to "sensitive to"

16-8: "Disclosing" to "Tracing"

17-6: "which has been" to "which was"

17-14: move "in SELEN" to the end of the sentence

17-15: "increased physical realism" is a little odd

Figure 7: label 'x' and 'y' in the right-hand panel.

---

## Referee Comment (RC3) · Geruo A (Referee) · 5 Sep 2019

The manuscript describes an updated numerical model that solves the sea level equation. Compared with its previous iterations, the new model now accounts for shoreline migration and rotational feedback, and it features enhanced portability and computational efficiency. The theory session is generally easy to follow. The result session provides a clear overview of the model configuration (i.e. ice and Earth model inputs) and highlights the newly implemented features (i.e. shoreline migration and rotational effects). The manuscript fits the scope of the journal and it is generally well written. I

have a few minor comments listed as follows.

The authors first define the sea level as B = -T (Eq. 3 and Page 3, Line 21), and later express it in Eq. 16 as the difference between the sea surface height and the height of the solid Earth surface. I think Eq. 16 is a more intuitive definition of the sea level. Based on this equation, it is also straightforward to define topography as T = -B. I suggest the authors introducing Eq. 16 before Eq. 2. Following this definition, the ocean function can also be defined immediately (O = 1 when B>0 with no grounded ice).

Figure 8. Please consider increasing the range of the color scale to make the plot less saturated.

Page 15, Line 15 to the end of session 3.3.2. It would be helpful to clarify the typical range of errors of tide gauge measurement. I think this would help the readers understand the significance of the difference among the reported model runs.

Table 5. The label and the caption of Column (e) should be ICE-6G instead of ICE-5G.

Page 16, Line 1. Please clarify how the coherence between the two predictions is quantified.

Page 16, Line 20. It would be helpful to clarify that the "direct effect" is associated with the change in centrifugal potential. This would help the readers follow the discussion at Lines 21-25.

Page 17, Lines 19-25. This part of the conclusion focuses on the computational aspect of the model while the current result session is not organized in a way to highlight this aspect. It would be helpful to include a brief summary in the result session to justify this part of the conclusion, especially regarding the second point at Line 21.

---

## Author Comment (AC3) · 20 Sep 2019

**Comments by Reviewer: Volker Klemann**

General comments

**Point R1-0**
The authors present in this manuscript an update of numerical code which enables to calculate the gravitationally consistent interaction between a surface mass load and water mass load which compensates the total mass change. Previous versions of this code were presented already on a number of workshops dealing with glacial isostatic adjustment, GIA. In contrast to these versions, the authors improved the code in a number of aspects which are currently discussed in GIA, and extended the portability of the code a lot. These improvements, to my point of view justify a new publication in a method and code-oriented Journal like GMD.

As I don't understand this review as a testing of the code, I will focus my review purely on the presented manuscript, and will not consider the supplement in this regard, especially for the derivation. In general, the manuscript is clearly written, the focus lies on the considered theory which the authors present in the theory section, and, as an application, the authors chose a published and established forcing for which they discuss the output in detail. Due to the fact, that the manuscript presents a methodologically oriented study, they do not discuss deviations from their results to those presented in the original publication of this forcing. From my point of view, this is reasonable strategy.

R. We thank Volker Klemann for his positive evaluation and for the suggestions made. We have made efforts to address all his comments; when not, a justification is given. See the details given below.

**Point R1-1**
As a validation of the method they refer to the benchmark study Martinec et al., 2018, which was recently published, and to which they contributed with a preliminary version of this code version. Of course the question may arise, why the authors did not present the results for the benchmark, and discuss the addition of the rotational effect which was not considered in that benchmark this would be a nice extension to that study. If they won't do this, at least they should state, that the results of that version are identical with the current one, if rotation is switched off. Consulting that study, the SELEN code deviates a bit more than the order models when considering moving coastlines and floating ice. But this is only a suggestion, to establish more a benchmark study, than the quite complicated ICE6G vm5a applied here.

R. The benchmark study in Martinec et al., 2018 was conducted between codes that do not account for rotational feedback (or in which these effects have been switched off). While we certainly could recompute benchmark results taking into account rotational effects, they cannot be directly compared with other results; actually, we could only discuss the difference between SELEN results with and without rotation. To some extent, this is actually done in the manuscript; moreover, in a recently published review paper (Spada and Melini, Water, 2019) the effect of rotational feedback on GIA fingerprints has been discussed in greater detail. In the revised manuscript, we now refer explicitly the reader to this paper. We think that a benchmark between SLE solvers that include rotational feedback would be of great interest, however at this stage we believe that recomputing the 2018 benchmark results with the inclusion of rotational feedback would not add insight to the discussion. As suggested by the Reviewer, we explicitly remark in the revised manuscript that the SELEN4 results, without rotation and in the same conditions of the 2018 benchmark, coincide with the results published in that context. See also point **R2-1**.

In the following, I focus on individual aspects of the presentation.

to Introduction

**Point R1-2**

P. 2, L 16: Elasticity is material law, but not a rheology.

The reviewer is right, we have rephrased and we do not use "elastic rheology" in the revised paper.

to Theory

**Point R1-3**

To present a reduced version of a derivation is every time dangerous, especially if this is not put into an appendix but a supplement which is not part of the manuscript. So, a number of questions arose, which partly come up in the following comments.

We are aware that balancing detail and conciseness in a theoretical section is not an easy task. We thank the reviewer for his suggestions on this point. We agree that the supplement is not strictly part of the manuscript, but it is nevertheless available to the reader after publication of the manuscript.

**Point R1-4**

P. 4, L 7: You refer to SSM19, here you should at least specify on which principles this expression (Eq. 2) is based.

We have introduced an explicit expression for the load (L) in terms of mass per unit area, which should illustrate better the physical meaning of L. Eq. (3) (formerly Eq. 2) is now better framed since we have made explicit the ocean function, following the suggestion in **R1-5**.

**Point R1-5**

P. 4, L 10: The meaning of OF should be defined explicitly.

This has now been done, since it effectively helps a lot the understanding of our reasoning. See also **Point R1-6.**

**Point R1-6**

P. 4, L 12: You introduce here the term bedrock topography, but do not specify what it means, especially as you further down use this quantity, to derive changes in sea level.

This section of the manuscript has been reorganised a bit, in order to define topography (T) in terms of sea level (B), which is a more intuitive definition. We refer more explicitly to Kendall et

al. (2005), who follow the same definition on topography. We also define the ocean function explicitly (see also **Point R1-5**), which should help to better understand our reasoning. See also **Points R2-7 and R3-1.** Due to these rearrangements, we have modified the beginning of Section 2.2. accordingly.

**Point R1-7**
P. 4, L 16: Here and in the following, you use the 'cal' symbol to specify variations with respect to a reference state. If so, you can of course reduce the number of equations, e.g., Eq.s 6, 11, 13 and 15 become redundant. Furthermore, you do not speficy the reference state itself.

We understand the point, but we prefer to keep these equations as they stand, since they define very fundamental quantities that should be 'lost' if embedded in the text. We are now more explicit about the meaning of the reference state, also quoting the paper of Kendall et al. (2005), who uses the same approach to the SLE.

**Point R1-8**
P. 4, L 24: I don't htink that you have to refere to SSM19 to introduce the definition (7), but simply if follows from $\cal M / \int_e dA$.

We agree. The text has been modified accordingly.

**Point R1-9**
P. 5, L 2: This is somehow abrupt, and do you really need it here?

No, we do not need it here. We have removed the equation and a few lines of text.

**Point R1-10**
P. 5, L 4: What do you understand as a plausible surface load. At least you should guide the reader a bit more than referring to Bevis et al. 2016.

As we state, for plausible surface loads we mean surface loads that conserve the mass of ice+water. We better focus on Bevis et al. (2016) realization of plausible loads, by noting that the SLE is not relying upon a-priori definitions of the load distributed over the oceans, acting to compensate the ice load.

**Point R1-11**
P. 5, L13: Is 'stem' used here synonymously to 'is associated'? If so, why then using a different word.

We use 'associated'.

**Point R1-12**

P. 5, L 5ff: How do you ensure in (Eq. 9) that you considered all terms with respect to $\cal M$? In Eq. 5, I ike that \cal O can be one of \{-1, 0, 1\}\.

In (previous) Eq. (9) we are confident we have included all the terms wrt $\cal M$, based on the analysis we perform in SSM19. We are not suspecting that anything is missing. Probably here the reviewer is referring, in his second observation, to Eq. 15 (not 5). In this case, we agree, ${\cal O}$ can (only) take the values -1, 0, and +1. We note that, in the revised manuscript.

**Point R1-13**

P. 6, L 20ff: The definition of seal level from topography is a bit tricky to understand, as you did not define topography itself properly, also sea surface is not defined her properly. So, why not motivate it from Gauss definition of the geoid as that equipotential surface to which a static ocean surface would adjust? Then, the jump from N to r^ss is not as large. Also at the beginning you should specify what the sea level equation expresses. Only then you can obtain its most basic form as U-N. Also the relation to the water column might help the reader to understand its relation to loading more intuitively.

In **Point R1-6**, we have responded to the issue about the definition of topography, also with the aim of responding to similar points of **R2** and **R3**. In the revised manuscript, we have made a short and clear statement about the meaning of the SLE at the very beginning of this Section. We also make a reference to the water column just after having written the basic form of the SLE, to help intuition.

**Point R1-14**

P. 6, L 12: This is also known as Bruhns formula. As $\cal .$ is not applied to $Phi$, I suggest to write this equation a bit different: \cal G = \frac{\Phi - \Phi}{g_0}, to remain in the used schema of formula symbols.

Yes, we use this notation, now, here and elsewhere. We quote the Bruns formula.

**Point R1-15**

P. 6, L 13: One important reference in your discussion of the SLE is missing which is Martinec and Hagedoorn, 2014, who discuss in detail the SLE of a rotating body. They, for instance, distinguish between gravity and gravitational potential, where the latter only considers the surface loading, i.e. variation due to surface mass change and solid earth deformation. The

gravity potential, MH14 use for gravitational potential plus rotational effects. This is in accordance to Helmut Moritz, 1990, The figure of the earth.
So I would use gravity potential instead of geopotential.

Yes, we agree on all these points. We use the same terminology as in MH14, and we quote this important reference.

**Point R1-16**
P. 6, L 14: The meaning of c only depending on $t$, can easily be explained simply by the fact, that the displacement of the reference potential surface, N_0, does not represent the potential level to which the water surface will adjust. This is not only due to mass conservation but also due to the changes in the ocean basin. So, it is a bit more complicate.

We believe that a nice and simple explanation of the existence and of the meaning of the 'c constant' is given by Tamisiea (GJI, 2011), to which now we refer more explicitly.

**Point R1-17**
P. 7, L 10: Here and in the following, $O$ is dangerous to use, as it is easily mistaken for the reference state 0.

It should not be the case, since the reference state label "0" is used as a subscript, while "o" in this equation and in the following is a superscript.

**Point R1-18**
P. 7, L 12: From type setting it is nicer to use text abbrevations in formulas with an unslanted font, e.g., ${\cal S}^\text{ave}$.

This is somewhat subjective, and since **R2** and **R3** did not make a similar observation, we keep all the equations as they stand.

**Point R1-19**
P. 7, 14: For the definition of the different sea-levels see below. For me, equivalent is more a renormed quantity, but here you are using it with respect to a model dependent quantity, depending on the current ocean area Aˆo (\gamma, t) and \mu (gamma, t).
Both terms depend on the $\cal S$ and $\cal Lˆ{[abc]}$.

This is another nice observation. We prefer to keep the abbreviation "equ" but it is clear that something must be said about the fact that S^"equ" is associated to model-dependent "dynamic"

quantities, like A^o and \cal O. We now mention this important point explicitly in the revised version of the paper.

**Point R1-20**
P. 7, L 8ff: This derivation is quite interesting, and I have needed some time to understand this alternative expression, especially why in Eq. 33 $T_0$ appears instead of $T$. May be, a specification what 'using for $c$ the expression found in SSM19' does mean, might help, as c does not appear anymore. The interesting aspect is that this form is independent of the syphoning, which dominates <S>ˆe, and is put into <G-U>ˆO. May be this might help to understand Table 4 a bit more. So I guess the expression, $- <G-U>ˆO + Sˆave = c$ is missing.

For that part of the question that concerns "c", we mean that to obtain eq. (30) of the submitted manuscript we have used the expression obtained for c by the imposition of mass conservation, given in the supplement. We are now more explicit on the equation used for c, to avoid any ambiguities. We are not sure we have captured the meaning of the other issues raised in this point. In particular, we do not understand why the syphoning is mentioned in the Reviewer' comment. So we ask for a clarification, if possible.

**Point R1-21**
P. 8 L 3: Wikipedia defines the 'eustatic change' as the alteration of sea level due to changes in volume of water or ice, and also due to changes in ocean topography, which goes back to Suess. The article to my point of view is based also on Rovere et al. (2016, https://doi.org/10.1007/s40641-016-0045-7), that is, also steric changes are considered. I would call Eq. 36, the equivalent sea level, as you also do at P. 10, L. 14ff, expressing an ocean equivalent of the ice, which is simply a renormation of water mass expressed as water height. Sˆave, I like. Your Sˆeus, I would call \bar{\cal S}ˆice, what is the equivalent of ice at current state. Your Sˆequ is a help quantity, I would keep its definition. FC76 phrase Eq. (36) to be the eustatic sea level although the ocean basin might slightly change. The authors are aware of the fact and "tried to avoid [. . .] the word (eustasy) has received so many qualifications since it was introduced by Suess".
So, I would suggest not to use eustasy at all. It is a deprecated definition of sea level and mainly leads to misunderstanding.

We have re-worded and discussed in more detail the meaning of the definitions given, also quoting in particular the recent paper of Gregory et al. (2019) about the terminology to be used in the context of sea level change. We agree that misunderstanding could easily arise in this context, we hope the modified text can be considered satisfactory. We keep "eus" but we better specify his meaning, quoting Gregory et al. (2019) about the new term "barystatic", which should be preferred to "eustatic".

**Point R1-22**

P. 8, L 17: 'essentially' is a bit enigmatic, do you mean terms of d/o = 2/0 and 0/0 can be neglected as they are too small? See also MH2014.

Yes, it is what we mean. We revised the text to make this point clearer.

**Point R1-23**

P. 8. L 7: From here to the end of this section, you discuss the numerical implementation into SELEN. As the whole article is about SELEN, I would expect a bit more detail, how the equations are solved also with respect to the iterations you discuss in the following. So, this part I suggest to extend.

No, this part is not meant to discuss the numerical implementation into SELEN. As quoted in the text, details are given in SSM19 and we do not intend to duplicate that material here. We add a few lines, however, in which we briefly describe in a qualitative way how the solution is approached.

to A test run of SELEN

**Point R1-24**

In the introduction to this section you should mention that you also consider changes in rotation in addition to N, U and S.

Yes, right. We have done that.

**Point R1-25**

P. 10, L. 23: Why do you use an interpolation. To go from a fine to a coarse grid by interpolation is everytime a bit dangerous. I would expect here a filtering or a binning algorithm.

This is an important observation and we thank the Reviewer for pointing it out. Indeed, we obtained the pixelized topography through an interpolation by means of the GMT 'grdtrack' module. However, in our numerical experiments, we used also binned topographies obtained through a sequence of 'blockmean' and 'grdtrack' GMT modules. The two realizations of topography turn out to be different in regions where small-scale relief features are present. However, we verified that all the relevant numerical results do not change within the numerical precision we used in the manuscript. In the revised text, we explicitly warn the user that for regional analyses focused on areas with small-scale topographic features, particular care shall be devoted to the realization of topography on the Tegmark grid.

**Point R1-26**

P. 11, L 10: 'Not agreed results on ve. Love numbers'. Do you have a reference for this statement, and why this is an argument to neglect compressibility?

We have rephrased (in LATEX) as <<\textbf{Since we \marginpar{R1-26} are not aware of published, community-agreed sets of Love numbers for a multi-layered compressible viscoelastic model}, in the test run we rest on \textbf{an incompressible profile, for which agreed results have been obtained \citep{spada2011benchmark}.}>> This is not an argument to neglect compressibility, it is just a remark. In the case Dr. Klemann (**R1)** can provide a reference contradicting this remark, we would change this statement accordingly. Beside this, SELEN4 can of course be used with any set of Love numbers, either community agreed or not agreed.

**Point R1-27**

P. 11, L 15: I would simply start the sentence: 'Numerical values of [. . .] in Table 3 and its caption, respectively', to avoid using two times reference, and the reader to search for the tidal love numbers in the table rows.

We agree; these changes have been done.

**Point R1-28**

P. 11, L 17 and P. 14, L 9: For the captions use 'Glacial isostatic adjustment in the past' and '[...] at present day' or alternatively 'Paleo glacial isostatic adjustment' and 'Present-day [...]'.

We use "GIA at present" as a title for Section 3.3 and "GIA in the past" for Section 3.2.

**Point R1-29**

P 11, P 27: I would in one sentence introduce the three configurations you discuss in the following.

We have been more specific and careful to describe the three configurations.

**Point R1-30**

P. 13, L 2: Why do you mention the small region around Patagonia with a reference only. Why not citing Klemann et al. (2015, DOI 10.1007/s41063-015-0004-x) for the East Siberian shelf or Lambeck for the Sunda Shelf. Furthermore, the light blue areas may represent locations covered with grounded ice, but also floating ice might be present there. The ice extent is not shown in Fig. 5.

We are aware that Fig. 5 is not including the ice. This is clearly stated in the second paragraph of 3.2.2. Paleo-maps also showing the ice sheets shall be included in future releases of SELEN4. The reviewer is right, some important references were missing; now we quote Klemann et al and some others with reference to GIA modeling in specific areas. SELEN can visualize the Ocean Function, in addition to paleotopography. OF maps are located in the /OFU output folder, also showing the distribution of the floating and grounded ice at every time increment. To limit the length of the main text, we do not show them here, where we focus only on topography (folder /TOP). However we now mention them, so the reader is aware of these maps in SELEN. The whole paragraph has been somewhat rearranged to fit the requirements of **R1**.

**Point R1-31**
P. 13, L 20. As stated before the MH14 paper, is in my point of view an important contribution to the discussion of rotation in GIA.

We agree and we quote the MH14 paper also in this context.

**Point R1-32**
P. 13, L 25: 'since' used twice.

Right. We use 'because' the second time, instead of 'since'.

**Point R1-33**
P. 14, L 8: For such statement, you should at least relate the order to that of other transient processes observed or proposed for rotational variations.

It is unclear to us, what the Reviewer is pointing to, here. We did not change the text.

**Point R1-34**
P. 14, L 11: Why 'shall', although you consider it.

Indeed. We now simply write "we consider".

**Point R1-35**
P. 14, L 15: Here you define $\cal S$ as relative sea level, I suggest you to use this definition from the beginning, what also would clearify what the SLE is solving for.

We define $\cal S$ as "relative sea level change", uniformly throughout the manuscript.

**Point R1-36**

P. 14, L 16: would repeat 'frame' also before b, c and d.

Done**.**

**Point R1-37**

P. 15, L 4: This is only clear from the fact that the syphoning effect is inside the definition of $\gamma$-depending part, otherwise the reader would wonder.

We are not sure we understand this comment. We did not change the text.

**Point R1-38**

P. 15, L 8: Instead of 'decontamination', geodesists speak about 'correction for' an effect.

We mention both terms.

**Point R1-39**

P 15, L 20: 'then' to 'them'.

OK.

**Point R1-40**

P. 16, L 23–25: This is an interesting aspect. Can you give a formula to indicate which term has to neglected as direct rotational effect?

We now refer explicitly to Eq. S173 of the supplement, showing the 1+k structure that we quote in the main text. We do not think it is necessary to duplicate it here.

**Point R1-41**

P. 16, L 32: Is this the degree variance, and why not presenting the equation here?

Because there is no necessity to duplicate Eq. S477 here, in our opinion.

to Conclusions

**Point R1-42**

P. 16, L 23–25: This is an interesting aspect. Can you give a formula to indicate which term has to neglected as direct rotational effect?

This point has been made above already, see **R1-40.**

**Point R1-43**

P. 16, L 32: Is this the degree variance, and why not presenting the equation here?

This point has been made above already, see **R1-41.**

**Point R1-44**

P. 17, L 12: replace 'runs' by 'run'.

Yes, done.

**Point R1-45**

P. 17, L 15: Not sure, if one can speak about 'physical realism'.

Right. Only realism.

to figures and tables

**Point R1-46**

Figure 1: In this color scheme, the dark ice thicknesses make it difficult to identify the coast lines. In this caption and in the further ones, I would not state which plotting command you used.

Right. We now use a red contour for the coastlines. We prefer to leave the command we use, for the sake of ease of reproducibility.

**Point R1-47**

Figure 2: I would skip this figure.

We have been tempted to skip this figure, too. But after consideration that it represents an important quantity, i.e., the final condition of the SLE, we would prefer to leave it.

**Point R1-48**

Table 2: 'Density, [...] values specifying the parameters of the homogeneous layers defining the adopted [...]'. 'Some spectral [...]' I would skip this sentence, as it does not contribute to the table content.

We have simplified a bit the second sentence, which was effectively too long.

**Point R1-49**

Table 3: The PMTF is not defined in the manuscript. Without it, the Aˆ[es] and [hkl]ˆT are not necessary. In this context it would be interesting, if you need also viscoelastic tidal love numbers.

We agree. We have skipped the PMTF information. We have left the tidal Love numbers information, as suggested by the Reviewer.

**Point R1-50**

Figure 8: Can you specify the extreme values reached in the respective plots?

Done.

**Point R1-51**

Table 4: I would skip the integrals over the whole earth, as these are clear from the definition of U and G.

We know that these are clear, but it is a nice indication of the precision of the SELEN4 numerical results. We prefer not to remove them.

---

## Author Comment (AC4) · 20 Sep 2019

**Comments by Reviewer: Samuel Kachuck**

**Point R2-0**

This manuscript details the background and significant improvements to an opensource implementation of the sea-level equation. As solutions to this equation are essential to understanding relative sea level and geodetic evolution, the open-source method and documentation are welcome and the changes - including the physics of rotation and shoreline migration and the modularization of the code - warrant publication.

In addition, I believe the inclusion of the derivation to be useful to the community. Pursuant to that, there are areas where the derivation could be made even clearer, as noted below. Occasionally, a proliferation of variable definitions and abbreviations clouds the exposition, and a simplification would be welcome.

We thank Samuel Kachuk for his positive evaluation and for the suggestions made. We have made efforts to address all his comments; when not, a justification is given. See the details given below.

**Point R2-1**

Finally, as the authors have just used this code's predecessor to participate in the benchmarking exercises of Martinec, et al. 2018, I think updating the contributed results with the new capabilities is warranted. In particular benchmark E, in which it seems they did not participate. As this is perhaps a more technical detail, I think it should be satisfactory to put this in the supplement, with perhaps a comment in the introduction to section 3. To this end, it would also be very interesting to see some discussion of how the included physics of rotation affect the results of those benchmarking cases.

A prototype version of SELEN4 has been used to compute results for the Martinec et al., 2018 exercise in all the considered benchmarks, including benchmark E (see Table 6 in Martinec et al., 2018). As suggested by R1, we explicitly state in the revised manuscript that the version of SELEN4 that we are publishing, when configured without rotation and in the same conditions of the 2018 benchmark, give numerical results that coincide with those published in that context. We believe that recomputing benchmark results with rotational effects would not add insight to the discussion, since all the 2018 exercises assumed no rotational feedback and therefore we cannot use them to validate our approach to rotational feedback modeling (of course, a new benchmarking initiative with a specific focus on the rotational effects would be of great interest). A discussion the difference between different rotation theories on the true polar wander is given in the manuscript, while for a more detailed analysis of the impact of rotational feedback on GIA fingerprints we now refer the reader to a recently published paper (Spada and Melini, Water, 2019). See also response to point **R1-1**.

All further comments are presented in page-line format

Specific Comments

**Point R2-3**

1-15: I would like to mention that the python module, giapy, which was benchmarked in the Martinec, et al. 2018 paper (including shoreline migration and iteratively determined initial topography, though without rotation) is available for download, opensource (from https://github.com/skachuck/giapy), and similarly uses any normal-mode form love numbers. It is safe to say, however, that it has not yet been as comprehensively documented in its open-source release as SELEN.

We have modified the text accordingly, quoting giapy, in the abstract and in the introduction.

**Point R2-4**

2-15: "viscoelastic rheology...properly taken into account" see note to 1-15

Agree; see also point **R2-3**. We have been more specific, here.

**Point R2-5**

4-12: Could you use $S$ instead of $B$ to keep it all consistent?

No, we cannot, because we use S (in \cal style) to denote the relative sea level change. B has a sense, here, because it recalls the term 'bathymetry'.

**Point R2-6**

4-2: "ice and by the water in the ocean" refers to ice on the continent and water in the ocean, which is the basis for equation (2). Equation (1) refers to ice and water over the surface of the earth, i.e., $_e$.

We are not sure we can capture this point. Are the two equations in contradiction?

**Point R2-7**

4-13: given the detail of the surrounding derivation, could you spare a few words giving a precise definition of "topography"?

We define topography (T) in terms of sea level (B), which is a more intuitive definition. See alo points **R1-6** and **R3-1**, who have suggested a similar improvement.

**Point R2-8**

5-4: The meaning of "plausible" is not addressed in the referred paper beyond setting the integrated mass load variation, or equivalently the zero-degree coefficient, to zero.

A similar point has been made also by **R1** (see **R1-9** and **R1-10**). To respond to both, we have rephrased this part of the manuscript, and we are now more specific on the meaning of 'plausible load' and on the contents of the Bevis et al. paper. Thanks to both Reviewers for rising this issue.

**Point R2-9**

5-6: Could more interpretable labels be given to the components of the load variation than a, b, and c?

We have been thinking a lot to this opportunity, but we did not have any valid idea. Consider that labels (a,b,c) affect a number of other variables as the reviewer can see in the Supplement. And also the variable names that we use in the source code. Any suggestion about a more intelligent labeling is welcome.

**Point R2-10**

5-10: The repeated definition of the variation variables gets confusing. Could the font used throughout be defined as the variation with respect to some reference level?

On the contrary, we think that it becomes more clear, although redundant. We believe that we cannot omit these fundamental variables. See also our response to Point **R1-7.** To help the readers, we now say that we use calligraphic capital letters to denote variations of the corresponding fields.

**Point R2-11**

5-19: Can you give a physical meaning to the height variable $Q$? "Auxialliary variable" and "arbitrary reference density" are a little confusing and undermotivated. If you selected $\rho^r = \rho^w$, then it takes the form of a free water column, positive where there is liquid water, negative where the water is frozen, and zero otherwise.

The Reviewer is right, in the sense that other choices could probably provide a physical interpretation, which however could be become too complicated and misleading. By Occam's Razor, we prefer to leave this definition, leaving a more in-depth analysis of the possible physical meanings to a future study or to the readers.

**Point R2-12**

7-9: Throughout, when referring to equations in the supplement, could you refer to specific equations?

Revising the paper, we have done this in a number of places to respond to **R1**. These equations/sections are marked by Sx or Sx.y.

**Point R2-13**

9-12: Could you briefly describe the significance of the external and internal iterations?

The same was essentially requested in **R2-23**, to which we have responded by adding a new short paragraph in which we summarise the meaning of external and internal iterations.

**Point R2-14**
16-12: You could cite somewhere the work done by Barletta and Bordoni (2013) on the effects of implementation of ice histories.

We would be happy to cite it, but we do not see in which part of our manuscript this could be useful. Any suggestions?

**Point R2-15**
17-27: see note to 1-15.

Yes, we agree. The text has been modified accordingly.

**Point R2-16**
Table 2: For clarity, you might articulate that these are piecewise constant layers whose upper radii are given by the variable $r$.

We have modified the style of the Table, giving both the lower and the upper radius, which makes things clearer, in our opinion. Note that the rigidities in the lower mantle where erroneously multiplied by a spurious factor of 10. This has been fixed.

**Point R2-17**
Figure 4: Is the ordering significant? If so, state what guides it. The non-numerical ordering makes it difficult to locate them from the text quickly.

The reviewer is definitively right here. Form the numbers, one cannot 'see' where the location is. We have added a map to help the reader. Thanks.

**Point R2-18**
Figures 5 and 6: the color scale looks different between these figures for the resolutions in the manuscript draft. Make sure to check this as it goes through editorial.

Thanks, we shall pay attention to this.

Technical corrections

**Point R2-19**
2-2: "available since" is a bit awkward, consider "dating from"

OK.

**Point R2-20**

2-10: "Despite GIA is now tightly" missing clause between "Despite" and "GIA"

We have rephrased into "Despite the GIA phenomenon is now tightly..."; we hope that this can be considered correct.

**Point R2-21**

2-16: "limited the" missing word "to"

The whole paragraph has been rephrased in response to Reviewer **R1**.

**Point R2-22**

2-16: "elastic rheology" to "elastic deformation"

Yes, we have changed this, also in response to **R1-2.**

**Point R2-23**

2-20: "Love numbers" to "Love number"
OK.

**Point R2-24**

2-28: "is hosting" to "has hosted" for tense agreement

OK.

**Point R2-25**

2-30: "Since year" missing work "the"

OK.

**Point R2-26**

3-24: "taken by the SLE" to "the SLE takes"

OK.

**Point R2-27**

3-24: remove "an"

OK.

**Point R2-28**

6-6: Should $\hat{}{ss}$ be $\hat{}{se}$?

Yes, indeed. A Major typo that has been noted also by others.

**Point R2-29**

8-25: "accomplished projecting" missing word "by"

Yes.

**Point R2-30**

11-14: "with origin in the whole Earth's: : :" to "with the origin at the Earth's: : :"

Yes.

**Point R2-31**

12-7: "from high-resolution but also from" to "from both high-resolution and"

Yes.

**Point R2-32**

12-21: The first clause "Differently...program" is awkward to read and the sentence would be fine without it.

Agree, 1st clause removed.

**Point R2-33**

13-5: The meaning of the word "safely" is not clear.

Agree, word 'safely' removed.

**Point R2-34**

13-25: "since" used twice in one sentence.

Agree, one of the two 'since' has been substituted by 'because'. See also **R1-32**.

**Point R2-35**

13-27: Consider adding "and shown in Figure 7" at the end of that sentence for a smoother transition.

Agree on the smooth transition.

**Point R2-36**

13-30: The sentence beginning with "Further results" could be revised to "Dashed curves show results obtained with the traditional theory."

Much better.

**Point R2-37**

14-9: Inconsistent abbreviations in section headings; "GIA versus "Glaical Isostatic Adjustment"

See also point R1-28, we are now consistent with the use of <<GIA>> in the section headings.

**Point R2-38**

14-25, 14-31: "regardless the" missing word "of"

OK.

**Point R2-39**

15-3: "very affected by" to "sensitive to"

OK.

**Point R2-40**

16-8: "Disclosing" to "Tracing"

OK.

**Point R2-41**

17-6: "which has been" to "which was"

OK.

**Point R2-42**

17-14: move "in SELEN" to the end of the sentence

OK.

**Point R2-43**

17-15: "increased physical realism" is a little odd

Yes, we change into realism, see also **R1-45**.

**Point R2-44**

Figure 7: label 'x' and 'y' in the right-hand panel.

OK.

---

## Author Comment (AC5) · 20 Sep 2019

**Comments by Reviewer: Geruo A**

**Point R3-0**
The manuscript describes an updated numerical model that solves the sea level equation. Compared with its previous iterations, the new model now accounts for shoreline migration and rotational feedback, and it features enhanced portability and computational efficiency. The theory session is generally easy to follow. The result session provides a clear overview of the model configuration (i.e. ice and Earth model inputs) and highlights the newly implemented features (i.e. shoreline migration and rotational effects). The manuscript fits the scope of the journal and it is generally well written. I have a few minor comments listed as follows.

We thank Geruo A for his positive evaluation and for the suggestions made. We have made efforts to address all his comments; when not, a justification is given. See the details given below.

**Point R3-1**

The authors first define the sea level as B = -T (Eq. 3 and Page 3, Line 21), and later express it in Eq. 16 as the difference between the sea surface height and the height of the solid Earth surface. I think Eq. 16 is a more intuitive definition of the sea level. Based on this equation, it is also straightforward to define topography as T = -B. I suggest the authors introducing Eq. 16 before Eq. 2. Following this definition, the ocean function can also be defined immediately (O = 1 when B>0 with no grounded ice).

This section of the manuscript has been rephrased, in order to define topography (T) in terms of sea level (B), as also suggested by Reviewer 1.  See also Points **R1-6** and **R2-7.**

**Point R3-2**

Figure 8. Please consider increasing the range of the color scale to make the plot less saturated.

While the two top panels are a bit saturated, the two bottom ones are not (see min/max values in the caption). Using two different scales for the top and bottom hinder a easy intercomparison between the fingerprints, so we have decided to leave the figure as it stands**.**

**Point R3-3**

Page 15, Line 15 to the end of session 3.3.2. It would be helpful to clarify the typical range of errors of tide gauge measurement. I think this would help the readers understand the significance of the difference among the reported model runs.

This is a very useful suggestion, and we thank the Reviewer for that. We make this point when we compare the 'best' SELEN prediction (R100/L512/I5) to the original implementation of ICE-6G\_C(VM5a).

**Point R3-4**

Table 5. The label and the caption of Column (e) should be ICE-6G instead of ICE-5G.

Yes. We have made this change.

**Point R3-5**

Page 16, Line 1. Please clarify how the coherence between the two predictions is quantified.

For coherence we mean that the two predictions have the same sign. We avoid using this term and rephrase, now.

**Point R3-6**

Page 16, Line 20. It would be helpful to clarify that the "direct effect" is associated with the change in centrifugal potential. This would help the readers follow the discussion at Lines 21-25.

Right, OK.

**Point R3-7**

Page 17, Lines 19-25. This part of the conclusion focuses on the computational aspect of the model while the current result session is not organized in a way to highlight this aspect. It would be helpful to include a brief summary in the result session to justify this part of the conclusion, especially regarding the second point at Line 21.

In this part of the conclusions, we just want to briefly highlight the major improvements we implemented in SELEN4 from a technical and practical standpoint. We think that it is it not worth discussing technical aspects like code organization or customization of input files in the main paper, since these are illustrated in detail in the user guide. On the other hand, we agree with the Reviewer that our statement about code parallelism on line 21 needs some quantitative support, so now we explicitly refer to the supporting material, where the scaling of SELEN4 has been thoroughly characterized.